



# Interannual variability of the initiation of the phytoplankton growing period in two French coastal ecosystems

Coline Poppeschi[1], Guillaume Charria[1], Anne Daniel[2], Romaric Verney[3], Peggy Rimmelin-
Maury[4], Michaël Retho[5], Eric Goberville[6], Emilie Grossteffan[4], Martin Plus[2]

[1]Ifremer, Univ. Brest, CNRS, IRD, Laboratory for Ocean Physics and Satellite remote sensing (LOPS), IUEM,
29280 Brest, France.
[2]Ifremer, DYNECO, Pelagic Ecology Laboratory (PELAGOS), 29280 Brest, France.
[3]Ifremer, DYNECO, Hydrosedimentary Dynamics Laboratory (DHYSED), 29280 Brest, France.
[4]OSU-European University Institute of the Sea (IUEM), UMS3113, 29280 Plouzané, France.
[5]Ifremer, Morbihan-Pays de Loire Environment Resources Laboratory (LERMPL), 56100 Lorient, France.
[6]Unité Biologie des Organismes et Ecosystèmes Aquatiques (BOREA), Muséum National d'Histoire Naturelle,
CNRS, IRD, Sorbonne Université, Université de Caen Normandie, Université des Antilles, Paris, France

*Correspondence to*: Coline Poppeschi (coline.poppeschi@ifremer.fr)

**Abstract.** Decadal time series of chlorophyll-*a* concentrations sampled at high and low frequencies are explored
to study climate-induced changes on the processes inducing interannual variations in the Initiation of the
Phytoplankton Growing Period (IPGP) in early spring. In this study, we specifically detail the IPGP in two
contrasting French coastal ecosystems: the Bay of Brest and the Bay of Vilaine. A large interannual variability in
the IPGP is observed in both ecosystems in connection with variations of environmental drivers (solar radiation,
sea temperature, wind direction and intensity, precipitation, river flow, sea level, currents and turbidity). We show
that the IPGP is delayed by around 30 days in 2019 in comparison with 2010. The use of a one-dimensional vertical
model coupling hydrodynamics, biogeochemistry and sediment dynamics shows that the IPGP is generally
dependent on the interaction between several drivers. Interannual changes are therefore not associated with a
unique driver (such as increasing sea surface temperature). Extreme events also impact the IPGP. In both bays,
IPGP is sensitive to cold spells and flood events. The interannual variability of the IPGP is significant and strongly
conditioned, at the local scale, by a combination of several environmental parameters, with a larger sensitivity to
sea temperature and light conditions, linked to the turbidity of the system. While both bays are hydrodynamically
contrasted, the processes that modulate IPGP are similar.

**Keywords**
Phytoplankton biomass, Long-term *in situ* observations, Coastal ecosystems, Extreme events, Climate change.

## 1 Introduction

Although studied for 70 years (Sverdrup, 1953), the optimal conditions that trigger the Initiation of
Phytoplankton Growing Period (IPGP) in ocean waters in early spring are not well understood (Sathyendranath *et
al.,* 2015). Three main theories are proposed to date: the Critical Depth Hypothesis (Sverdrup, 1953), the Critical
Turbulence Hypothesis (Huisman *et al.,* 1999) and the Disturbance-Recovery Hypothesis (Banse, 1994;
Behrenfeld, 2010; Behrenfeld *et al.,* 2013). These hypotheses, determined with specific scales and ecosystems, are
still regularly debated owing to the use of more efficient models and new observation systems that allow the
collection of large *in situ* datasets (Boss and Behrenfeld, 2010; Rumyantseva *et al.,* 2019; Caracciolo *et al.,* 2021).
Moreover, theories proposed for the open oceans are not relevant in coastal zones. Although the IPGP is also
determined by a combination of factors including the level of light available for phytoplankton, the availability of
dissolved inorganic nutrients, the water residence time and the grazing by zooplankton, coastal waters remains
highly dynamic and productive ecosystems at the interface between land and sea (*e.g* Gohin *et al.,* 2019; Liu *et
al.,* 2019). Because coastal systems are directly influenced by anthropogenic inputs from rivers, no nutrient
limitation is observed in late winter. A myriad of factors and mechanisms can affect the IPGP on coastal areas
(Townsend *et al.,* 1994; Cloern, 1996), with the incident light at the air/sea interface on top of the ocean (Glé *et
al.,* 2007) and sea surface temperature (Trombetta *et al.,* 2019) in late winter being the main forcings. Low water
turbidity also plays a major role and allow deeper light penetration (Iriarte and Purdie, 2004); this occurs by low





vertical mixing conditions, i.e. weak wind (Tian *et al.,* 2011), neap tide (Ragueneau *et al.,* 1996) and in absence of flooding events (Peierls *et al.,* 2012). Depending on the morphology and hydrodynamics of coastal zones (estuaries, bays, lagoons), the importance of controlling factors can be variable (Cloern, 1996). The variability of IPGP plays a major role on several biological compartments in coastal ecosystems: change in the timing of IPGP can impact zooplankton and fish by inducing species replacements (Sommer *et al.,* 2012), or phytoplankton itself by changing species composition or the succession of species (Ianson *et al.,* 2001; Edwards and Richardson, 2004;
Chivers *et al.,* 2020).

By amplifying or modifying environmental forcings, it is now well-documented that global climate change may influence the IPGP in coastal areas (Smetacek and Cloern, 2008; Barbosa *et al.,* 2010; Pearl *et al.,* 2014; IPCC, 2021). Heat waves, as opposed to cold spells, have become more frequent in recent years and can advance or delay the IPGP respectively (Gomez and Souissi, 2008). Wind storms, by inducing vertical mixing and
sediment resuspension, can have a significant effect on water turbidity which in turn limits light penetration and therefore influences the IPGP. Floods, following heavier rainfall, may increase continental erosion, land-based transfers and ultimately nutrient inputs to coastal ecosystems. Because coastal ecosystems are strongly influenced by changes in land use, detecting long-term climate-induced signals is challenging (Krompkamp and Van Engeland, 2010).

Our study is based on two geographically close but hydrodynamically different ecosystems: (1) the Bay of Brest, a shallow semi-enclosed bay with well-mixed waters (Le Pape and Menesguen, 1997) and (2) the Bay of Vilaine, a shallow open bay with long water residence times (Chapelle *et al.*, 1994). These two coastal ecosystems are strongly impacted by anthropogenic pressures, such as intensive agriculture (Ragueneau *et al.,* 2018; Ratmaya *et al.,* 2019). Most studies dealing with IPGP are mainly based on discrete water sampling (Iriarte *et al*., 2004;
Tian *et al.*, 2011) or modeling (Townsend *et al.,* 1994; Philippart *et al.,* 2010) and only a few investigated long-term high-frequency observations (Gomez and Souissi, 2008; Iriarte and Purdie, 2004) to assess interannual variability of the IGPG and to identify the triggering and controlling factors. Here, we develop a method to detect and analyze IPGP in coastal environments, combining high-frequency decadal *in situ* observations and modeling, using a 1DV hydro-sedimentary and biogeochemical coupled numerical model.

In this study, we aim to better understand interannual changes in the IPGP in the current context of global climate change over the last 20 years. We first detect and analyze the temporal variability of the IPGP and we then quantify how environmental forcings influence its dynamics. The potential impact of hydro-meteorological extreme events, such as cold waves, flood events and wind bursts, on the IPGP is then investigated.


## 2 Data and methods

### 2.1 Study areas

The study focuses on two northwestern French coastal ecosystems, the Bay of Brest and the Bay of Vilaine, which are both impacted by excessive nutrient inputs from watersheds, but exposed to different hydrodynamic conditions.

The Bay of Brest is a semi-enclosed bay (180 km$^2$) with 50% of the surface shallower than 5m depth. The
Bay is connected with the Atlantic Ocean (Iroise sea) through a narrow and shallow strait. Tidal variation reaches 8 m during spring tides, which represents an oscillating volume of 40 % of the high tide volume. Freshwater inputs are essentially from the Aulne river (catchment area 1875 km$^2$, mean river flow 26 m$^3$ s$^{-1}$), and also from two smaller rivers, the Elorn (catchment area 385 km$^2$, mean river flow 6 m$^3$ s$^{-1}$) and the Mignonne (catchment area 111 km$^2$, mean river flow 1.5 m$^3$ s$^{-1}$). Because of the macrotidal regime, the high nitrate concentrations do not
generate important green tides (Le Pape et al., 1997) and the strong decreases in the Si:N and Si:P ratios did not exhibit dramatic phytoplankton community shifts from diatoms to non-siliceous species in spring (Del Amo *et al*., 1997) according to the high Si recycling (Ragueneau *et al*., 2002; Beucher *et al*, 2004).

The Bay of Vilaine is a mesotidal open bay (69 km$^2$) under the influence of the Vilaine (catchment area
10 500 km$^2$, mean river flow 70 m$^3$ s$^{-1}$) and the Loire (catchment area 117 000 km$^2$, mean river flow 850 m$^3$ s$^{-1}$) river discharges, with tidal ranges varying between 4 and 6 m (Merceron, 1985). The Loire river plume tends to spread northwestward with a dilution of 20- to 100-fold by the time it reaches the Bay of Vilaine (Ménesguen *et al*., 2018). The Vilaine river plume tends to spread throughout the bay before moving westward (Chapelle *et al*., 1994). The water residence time varies seasonally between 10 and 20 days (Chapelle *et al*., 1994). The water
circulation is mainly driven by tides, winds and river flows (Lazure and Jegou, 1998). This bay is well known as one of the most sensitive European Atlantic coastal ecosystems to eutrophication (Ménesguen *et al*., 2019). The Bay of Vilaine has undergone eutrophication over recent decades mainly due to high nutrient inputs from the Vilaine and Loire rivers (Rossignol-Strick, 1985; Ratmaya *et al.*, 2019).



**2.2 *In situ* observations**


COAST-HF-Iroise (Rimmelin-Maury *et al*., 2020) and COAST-HF-Molit (Retho *et al*., 2020) are two high-frequency monitoring buoys of the French national observation network COAST-HF[1] (Répécaud *et al*., 2019; Farcy *et al*., 2019; Cocquempot *et al*., 2019; Poppeschi *et al*., 2021) located respectively in the Bay of Brest (4.582°W; 48.357°N) and in the Bay of Vilaine (2.660°W; 47.434°N) (Fig. 1). COAST-HF-Iroise has been

operating in the strait between the Bay of Brest and the Atlantic Ocean since 2000. COAST-HF-Molit buoy has been sampling the plume of the Vilaine river since 2008. Buoys are deployed during the whole year except for COAST-HF-Molit only available for part of the year prior to 2018 (from mid-February to early September, i.e. from day 50 to 250 for the period 2008-2017). Depending on the tide, the depth at the mooring sites ranges from 11 to 17 m for both COAST-HF buoys. Environmental parameters (temperature, salinity, turbidity, dissolved

oxygen and Chl-*a* fluorescence) are measured at 2 m (COAST-HF-Iroise) and 1.3 m (COAST-HF-Molit) below the surface, every 20 and 60 minutes. The Chl-*a* fluorescence is measured by a Turner CYCLOPS-7 Sensor (precision $\pm$ 5%) and is considered as a proxy of phytoplankton biomass (unit FFU).

Sub-surface Chl-*a* concentrations are provided from two French marine monitoring networks, the SOMLIT coastal observation network[2] and the REPHY (French Observation and Monitoring program for Phytoplankton

and Hydrology in coastal waters)[3]. They are collected bimonthly respectively at the SOMLIT-Brest (4.552°W; 48.358°N) and the REPHY-Loscolo (2.445°W; 47.496°N) stations which are close to the COAST-HF stations. Chlorophyll-*a* concentrations are measured with either spectrophotometric or fluorimetric methods (Aminot and Kérouel, 2004).

Daily river flows are measured at gauging stations (French hydrology "Banque Hydro" database[4]), located

close to the main river mouths [Aulne-Gouezec (4.093°W; 48.205°N), Loire-Montjean (1.78°W; 47.106°N)]. The Vilaine river flow is controlled by a dam, and data were provided by the Vilaine Public Territorial Basin Organization[5] (Fig. 1).

The tide gauge stations (Shom[7]) at Brest (4.495°W; 48.382°N) and Crouesty (2.895°W; 47.542°N) record the sea level every minute.

Precipitation, air temperature, wind direction and intensity, and the solar flux data are retrieved every 6 minutes from two meteorological stations from the Météo-France observation network[6]: Guipavas (4.410°W; 48.440°N) and Vannes-Séné (2.425°W; 47.362°N) (Fig. 1). The solar flux can be used here as a proxy for subsurface PAR (Photosynthetically Available Radiation).

**2.3 MARS3D-1DV modeling experiments**

**2.3.1 MARS3D-1DV model**

A 1DV (one-dimensional vertical) model configuration is implemented to simulate changes in biogeochemical variables due to hydrodynamics and sediment dynamics in both bays.

The hydrodynamical model is based on the MARS3D (3D hydrodynamics Model for Applications at Regional Scale) code (Lazure and Dumas, 2008). This model is a primitive equation model with a free surface and uses the Boussinesq and hydrostatic pressure assumptions. Here, we use the 1DV configuration of the model, with 10 vertical sigma levels for 15 m depth. The time step is 30 s.

The sediment model (MUSTANG - Le Hir *et al.,* 2011; Grasso *et al.,* 2015; Mengual *et al.,* 2017) is designed

to simulate the transport and changes in different sediment mixtures. In the sediment, 50 layers (refined near the surface) for a total thickness of 40 cm are implemented. Four sediment classes are considered: muds (diameter 10 $\mu$m), fine sand (diameter 100 $\mu$m), medium sand (diameter 200 $\mu$m) and coarse sand (diameter 400 $\mu$m). The sediment dynamics (transport in the water column, exchanges at the water/sediment interface, erosion/deposition

---

[1] www.coast-hf.fr, data available on www.coriolis-cotier.org

[2] https://somlit.fr

[3] https://doi.org/10.17882/47428

[4] www.hydro.eaufrance.fr/

[5] https://www.eptb-vilaine.fr/
[6] https://donneespubliques.meteofrance.fr/
[7] http://data.shom.fr





processes) are driven by an advection/dispersion equation for each sediment class (refer to Le Hir et al., 2011 for
a detailed description of the sediment model).

The biogeochemical model BLOOM (BiogeochemicaL cOastal Ocean Model) is derived from the ECO-MARS model (Cugier *et al.,* 2005; Ménesguen *et al.,* 2019) adding major processes of early diagenesis. Nitrogen, phosphorus, and silica cycles are studied considering four nutrients, respectively nitrate, ammonium, soluble reactive phosphorus, silicic acid (sorption/desorption of phosphate on suspended sediment and precipitation/dissolution of phosphate with iron processes are also included). The model is also represented by three phytoplankton classes (microphytoplankton, dinoflagellates, pico-nano-phytoplankton), two zooplankton classes (micro- and meso-zooplankton), and exchanges at the water/sediment interface and inside the sediment compartment.

### 2.3.2 MARS3D-1DV model sensitivity experiments

These three models (hydrodynamical, sediment and biogeochemical) are coupled online during simulations and allow the nutrient and phytoplankton dynamics in both bays to be reproduced. The simulation for the Bay of Brest does not include nutrient inputs from the sediment because it is considered to be negligible around the COAST-HF-Iroise station.

Dissolved and particulate variables are defined in the water column and in the sediment. Initial values for both bays are uniform over the initial vertical profile (Table 1) and are based on a 3D realistic coupled simulation during the year 2015 the 15[th] of February extracted at the position of COAST-HF-Iroise for the Bay of Brest and at the position of COAST-HF-Molit station for the Bay of Vilaine.

To evaluate the sensitivity of the biogeochemical dynamics to environmental conditions, sensitivity experiments are then performed using the coupled MARS3D/BLOOM/MUSTANG 1DV model configuration. All simulations are started at the end of winter (15[th] February) and run until the end of the year. The range of values used in the sensitivity experiments are derived from the minimum and maximum observed *in situ* data. Each parameter is tested with a constant value for the whole simulation.

Three parameters are individually explored in both bays:
- The air temperature in sensitivity experiments ranges from 4 to 14°C and is controlled by the intensity of solar radiations. Air temperature represents the main controlling parameter of Sea Surface Temperature in the 1DV model. This parameter drives the radiative fluxes in the model and then constrains the SST.
- Wind intensity effect on the IPGP is explored for values between 0 and 10 m s$^{-1}$. In the 1DV model, wind is a source of vertical mixing in the simulation.
- The Cloud Coverage (CC) sensitivity experiments ranged in value between 0 and 100% CC. This parameter is a driver of Photosynthetic Available Radiation (PAR) in the ocean. For the formulation of radiative fluxes in the 1DV MARS3D model, 100% cloud coverage allows an inflow of 38% of the total solar radiation in the water column. Each individual experiment is associated with a constant CC applied to the seasonal solar radiation.

As the sediment plays a role on the light penetration and acts as an active source of nutrients mainly in the Bay of Vilaine, the mud erosion rate (values between $2.10^{-5}$ and $2.10^{-7}$ kg m$^{-2}$ s$^{-1}$) is explored only in that bay (sand erosion rate fixed to 0.0001 kg m$^{-2}$ s$^{-1}$). For the sensitivity experiments, it drives a mass of sediment eroded and resuspended and a bottom input of nutrients in the water column.

A second set of experiments is conducted combining the effect of these environmental parameters in order to explore the cumulative or opposite effect on the IPGP. The upper and lower bounds of the range of environmental parameters are taken into account. Experiments are detailed in Table 5.

### 2.4 Data processing

#### 2.4.1 Chl-*a* fluorescence data

To analyze high-frequency time series of *in situ* Chl-*a* fluorescence, the Quenching effect (Lehmuskero *et al.*, 2018), a decrease in fluorescence in the presence of light (Fig. 2), is removed by analyzing only night-time data as reported in Carberry *et al.* (2019). Chl-*a* fluorescence data are studied on a daily basis, i.e. averaged from 10 pm to 5 am. Years with less than 75% of valid data (i.e. 2005, 2006, 2008, 2009 and 2018 in the Bay of Brest) are not considered.



### 2.4.2 Detection of the IPGP


We apply methods from the literature (Kromkamp *et al*., 2010; Philippart *et al*., 2010; Brody *et al*., 2013) to calculate annual IPGP values (not shown). Kromkamp *et al*. (2010) set an arbitrary beginning and end of the phytoplankton growing period at 20% and 80% of the cumulative Chl-a fluorescence measured from January 1[st] to December 31[st]. Similarly, Brody et al. (2013) consider a threshold of 5% above the yearly median chlorophyll.

Philippart *et al*. (2010) considers the beginning of the growing period as the maximum daily difference in Chl-*a* fluorescence.

Because we obtain unrealistic IPGP dates from our dataset when using the methods proposed by Kromkamp *et al*. (2010 - i.e. too late IPGP); Brody *et al*. (2013 - i.e. too early IPGP) and Philippart *et al*. (2010 - i.e. multiple IPGP), we propose an alternative detection method based on discontinuities of the Chl-*a* fluorescence signal (Fig.

3): daily FFU slopes are calculated based on a linear regression over a +/-2 day window for each day, from 1[st] January to 31[st] December, and each year. The IPGP date is identified when the slope exceeds a threshold value, defined as the median of the daily slopes, for the first time in the year. The end of the phytoplankton growing period is determined when the slope stabilizes below the threshold for at least 20 days for the last time in the year. The cumulative Chl-*a* fluorescence corresponds to the duration of the growing period.

### 225 2.4.3 Pattern of the phytoplankton growing period

The k-means method (Hartigan and Wong, 1979) is used to characterize the annual patterns of the phytoplankton growing period.

We exclude the year 2013 from the analysis of the Bay of Vilaine because of a large number of missing

data. When the interval over which consecutive data are missing is no longer than one week, we perform a linear interpolation to replace the missing data. A 5-day running average is applied to the Chl-*a* fluorescence signal and the data are then normalized by the maximum value. We analyze Chl-*a* fluorescence every year for 150 days after the IPGP.

Time series from both bays are merged before application of the k-means and the number of clusters (or

centroids) is set at 2 to distinguish the dominant patterns of the phytoplankton growth period at both sites. The use of a larger number of clusters is investigated and does not produce a pattern representing a large number of observed growing periods.

### 2.4.4 Detection of extreme events

The peak over threshold method (see Oliver *et al*., 2018 and Poppeschi *et al*., 2021 for further details) is used to detect hydro-meteorological extreme events such as cold waves, flood events and wind bursts. An event is considered as extreme if values are higher than a given statistical threshold for at least 3 consecutive days. In the present study, the 90-percentile threshold is selected to detect floods and wind bursts and the 10-percentile to detect cold waves. Seasonal anomalies are calculated over at least 20 years, by subtracting raw data from the winter

average value (cold spells) or from the spring average value (wind bursts and floods).

## 3. Results

### 3.1 Characterization of the phytoplankton growing period

The high-frequency Chl-*a* fluorescence time series at both sites show an intense seasonal cycle with low values from November to February and high values from March to October (Fig. 4). Focusing on the period from 2010 to 2019 in the Bay of Brest, the minimum *Chl-a* fluorescence is observed during the years 2012 and 2013 and does not exceed 7 FFU. In contrast, years such as 2010, 2014, 2015 or 2019 show Chl-*a* fluorescence values above 15 FFU but can be up to 20 FFU. In the Bay of Vilaine, a similar seasonal pattern is observed with higher

values reaching 50 FFU in 2013. Small (< 20 FFU) and high (> 35 FFU) Chl-*a* fluorescence amplitude are observed occasionally (in 2014 and 2017 and in 2013 and 2016, respectively). The Chl-*a* fluorescence is higher, almost double, in the Bay of Vilaine compared to the Bay of Brest with a mean cumulative Chl-*a* fluorescence around 580 FFU and 360 FFU, respectively (Table. 2). The high phytoplankton biomass of the Bay of Vilaine is corroborated by the concentrations measured by low-frequency observation programs (SOMLIT and REPHY).



260 The phytoplankton growing period ranges from approximately March 10th to September 30th in both regions (Table 2). The average duration of the phytoplankton growing period is 179 days in the Bay of Vilaine and 200 days in the Bay of Brest (Table 2). The phytoplankton growing period is characterized by successive blooms, whose number and intensity are variable from year to year (Fig. 4).

265 The main patterns of the phytoplankton growing period are identified by the two clusters (Fig. 5). Cluster 0 includes the phytoplankton growing period with two successive marked blooms in early spring and in summer, the intensity of the second bloom being highly variable. Cluster 1 is characterized by a plateau during the two first months of the phytoplankton growing period. Most of the patterns of the Bay of Vilaine are in cluster 0 while those of the Bay of Brest are in cluster 1 (Table 3). The years that stand out in the Bay of Brest (2002, 2010, 2014)
270 correspond to years with the highest cumulative *Chl-a* fluorescence ($\geq$ 450 FFU). The atypical years in the Bay of Vilaine (2011, 2017 and 2019) show the lowest cumulative *Chl-a* fluorescence ($\leq$ 450 FFU).

**3.2 Variability of the Initiation of the Phytoplankton Growing Period (IPGP)**

 Calculations performed to determine the IPGP for high- and low-frequency data yield comparable results (Fig.
275 6). The mean differences between the IPGP calculated with the high and low-frequency data are 5 and 8 days for the Bay of Brest and the Bay of Vilaine, respectively. A difference of only 4 and 6 days between the model simulations (reference year = 2015) and the high-frequency *in situ* data is observed in the Bay of Brest and the Bay of Vilaine, respectively.

280 A decadal variability of the IPGP is recorded from mid-February to mid-April in both ecosystems (day 50 to day 102 in the Bay of Brest and day 53 to day 93 in the Bay of Vilaine; Fig. 6). In the Bay of Brest, early IPGPs (day < 53) are observed in 2010 and 2013 whereas late IPGP (day > 93) are observed in 2001, 2017 and 2019. In the Bay of Vilaine, the earliest IPGP is detected in 2012 (day 53) and the latest in 2019 (day 93).

285 The variability of IPGP in the Bay of Brest shows two linear trends (Fig. 6a), with a decrease of 52 days from 2001 to 2010 (observed in both high- and low-frequency datasets), followed by an increase (+48 days) from 2011 to 2019, a decline also observed in the Bay of Vilaine (Fig. 6b). Over the period 2011-2019, the IPGP is shifted towards a later date by +3.5 days per year in the Bay of Vilaine and +3.7 days per year in the Bay of Brest.

**3.3 Analysis of environmental conditions driving the IPGP**

290 **3.3.1 Impact of environmental conditions on the IPGP**

 We next quantify the influence of environmental drivers on the date of IPGP (Fig. 7). These drivers represent the major limiting factors of the phytoplankton growth and comprise input of nutrients (river flow), PAR (incident light), Sea Surface Temperature - SST - (air temperature, incident light) and turbidity in the water column
295 (river flow, wind intensity, turbidity).

 The median values of the environmental drivers observed at the date of each annual IPGP are very close in both bays (Table 4) : temperate SST (10 °C), weak wind (3 m.s$^{-1}$), a medium PAR (1360 W m$^{-2}$), a low turbidity (7 NTU) and a weak sea level (1.6 m in the Bay of Brest and 0.9 m in the Bay of Vilaine). The IPGP occurs mainly
300 during neap tides, at 68 % in the Bay of Brest and in the Bay of Vilaine, respectively. The flow of rivers is lower during the IPGP with a flow of 46 m$^3$ s$^{-1}$ for the Aulne, 96 m$^3$ s$^{-1}$ for the Vilaine and 1196 m$^3$ s$^{-1}$ for the Loire.

 To assess how environmental drivers may impact (i.e. advance or delay) the IPGP, we focus on the 15
305 days before the mean day of the IPGP (day 68) and of each annual IPGP. The considered 15 days length is related to the typical water residence time in both bays (Frere *et al.*, 2017; Poppeschi *et al.*, 2021 for the Bay of Brest - Chapelle *et al.*, 1994; Ratmaya *et al.*, 2019 for the Bay of Vilaine). The earliest IPGP (IPGP < day 55), which occurred in 2010 (Fig. S1f) and 2013 (Fig. 7c) in the Bay of Brest and in 2012 (Fig. S2a) in the Bay of Vilaine, are associated with earlier occurrence of favorable conditions than the other years. Favorable conditions for IPGP
310 are also found early in 2002 (Fig. S1b) and 2016 (Fig. S1j) in the Bay of Brest, with an onset less than or equal to day 60.
The latest IPGP (IPGP > day 90), observed in 2001, 2003, 2017 and 2019 in the Bay of Brest (Fig. S1a,c,k,l) and in 2019 (Fig. S2g) in the Bay of Vilaine are associated with unfavorable environmental conditions until the date



315 of the IPGP. For example, the delay detected in 2017 in both bays is due to strong wind and a lack of PAR until the day of IPGP (Fig. S1k, Fig. S2e). A delay of the IPGP (> day 70) is also recorded in 2004, 2007 and 2012 (Fig. S1d,e,g) in the Bay of Brest, and in 2014 (Fig. 7d), 2017 and 2018 (Fig. S2e,f) in the Bay of Vilaine.

Finally, IPGPs start around day 68 ($\pm$3 days), on average, in 2011, 2014 and 2015 (Fig. 7a - Fig. S1h,i) in the Bay of Brest, and 2011, 2013, 2015 and 2016 (Fig. 7b - Fig. S2b,c,d) in the Bay of Vilaine. For example, in 2011 and 320 for both bays (Fig. 7a,b), low flow, wind and turbidity conditions are observed, so the IPGP requires only that the incident light increases in order to have a high enough temperature to trigger.

The interannual variability of the date of the IPGP is therefore not controlled by a unique environmental driver. When the values of the environmental drivers responsible for the IPGP (Table 4) are compared to the mean values of the environmental drivers 15 days before and after the IPGP (Table S1), threshold values are observed 325 in both bays: river flow is lower than usual (between 10 and 30 $m^3$ $s^{-1}$), temperature is close to the expected value (10°C), wind is weak (0.5 to 1.5 $m$ $s^{-1}$), PAR is stronger (>300 W $m^{-2}$), and turbidity is low (about 1.5 NTU).

### 3.3.2 Modeling the importance of the environmental drivers

The importance of each environmental driver on the IPGP is determined by MARS-1DV simulations starting 330 on February 1st (Fig. 8). Model results show that an early IPGP is associated with an initial air temperature (as a controlling driver in the model of the Sea Surface Temperature evolution) higher than 9 °C, resulting in a SST higher than 8 °C. Low wind intensity and weak Cloud Coverage (CC, as a PAR proxy) also faster the IPGP as well as a low turbidity (erosion rate, as a proxy).

The impact of the environmental drivers that advance or delay the IPGP are similar in both bays. Air 335 temperature has the potential to cause the greatest deviation from the mean IPGP, over 25 days in the Bay of Brest and 40 days in the Bay of Vilaine (Fig. 8). Wind, CC and turbidity have a lower impact on the IPGP (lower than 6 days in the Bay of Brest and 13 days in the Bay of Vilaine). In the Bay of Vilaine, the environmental drivers simulate larger delays of the IPGP than in the Bay of Brest.
340

In the Bay of Brest (Fig. 8a), only the air temperature variations (as a controlling proxy of the SST) have a real impact on the IGPG. If the air temperature does not exceed 8°C, the IPGP is not triggered before day 74 (Table 5, Exp 1), if the air temperature is above 13°C the IPGP starts on day 49 (Table 5, Exp 2). The variations of wind and CC induce weaker/lower shifts in the date of the IGPG, i.e. about one week at the most (Table 5, Exp 3,4,5,6).
345

In the Bay of Vilaine (Fig. 8b), the air temperature variations also have an important impact on the date of the IPGP. If temperature is equal or above 13°C the IPGP starts on day 45 (Table 5, Exp 2). If the erosion rate is only 2.10$^{-7}$ kg $m^{-2}$ $s^{-1}$, then the IPGP takes place only on day 76 (Table 5, Exp 7). If the air temperature is below 6°C then the IPGP is late and appears only after day 80 (Table 5, Exp 1). Similarly, if the erosion rate is 2.10$^{-5}$ kg $m^{-2}$ 350 $s^{-1}$, the IPGP does not occur until day 87 (Table 5, Exp 8).

However, very weak wind conditions (around 2 $m$ $s^{-1}$, Fig. S2a and S1f) could explain the very early IPGP observed in 2012 in the bay of Vilaine and in 2010 in the bay of Brest (respectively day 53 and day 50). For such conditions, the model advances the IPGP respectively on days 60 and 63, earlier than usual day 68 and day 69 355 (Fig. 8b).

From the MARS-1DV model simulations, the combined effect of the environmental drivers, namely air temperature, wind, CC and erosion rate (Fig. 9), can also be explored. The modeling conditions (hereafter called "Exp") are detailed in Table 5. There is a delay in both bays when the environmental parameters correspond to the 360 most extreme unfavorable combined IPGP values (temperature of 4°C, wind intensity of 10 m $s^{-1}$, CC of 100% and erosion rate of 2.10$^{-5}$ kg $m^{-2}$ $s^{-1}$ - Exp A). The IPGP then occurs 9 days later (i.e. twice as late as for any individual driver simulation) in the Bay of Brest and 64 days later in the Bay of Vilaine compared to the mean IPGP (day 68). The delays induced by the cumulative effects of the "temperature and wind" (Exp B) and the "temperature and CC" combinations (Exp C) are less important in the Bay of Brest than in the Bay of Vilaine (Fig. 365 9, Table 5, 9 and 5 days respectively). In contrast, no delay is observed for the combination "wind and CC" (Exp D) in the Bay of Brest as well as in the Bay of Vilaine with a minor impact of 6 days.

A combined effect that results in an earlier IPGP is simulated when conditions correspond to a temperature of 14°C, no wind intensity and CC, and an erosion rate of 2.10$^{-7}$ kg $m^{-2}$ $s^{-1}$ - Exp K. The early IPGP occurs also on the same day as other experiments with only two modified parameters such as Exp L and M (and even N for the 370 Bay of Vilaine). All the combined scenarios permit the occurrence of an earlier IPGP (by at least 5 additional days) compared to experiments that consider a single modified parameter.





This analysis enables environmental parameters to be classified with respect to their impact on the IPGP. In both bays, the temperature appears to be the key factor driving the IPGP. By combining the environmental drivers, the IPGP can occur even later or earlier than with a single forcing. In both bays, the combination of wind and CC has no impact on the IPGP, which occurs near the median day (Exp D and N). The extreme couplings of Exp A,E,F,G,J delay the date of IPGP later than detected in the observations for the Bay of Vilaine.

**3.4 Impact of extreme hydro-meteorological events on the IPGP**

**3.4.1 Cold spells**

The impact of cold spells on the IPGP is simulated with the MARS-1DV model based on two criteria: (i) the period of occurrence of the event, set in mid- or end February, (ii) the duration and intensity of the cold spell, which can be either short and weak (8 days, 7°C) or long and intense (20 days, 5°C) (Fig. 10).

In both bays, when the cold spell appears in mid-February, the IPGP is not impacted. However, it is delayed by about 15 days when occurring at the end of February. The duration of the cold spell, when longer than 15 days, also has an impact on the IPGP, with a delay of 13 and 12 days in the Bay of Brest and in the Bay of Vilaine, respectively.

Eight cold spells are detected in February in both bays between 2001 and 2019. In 2011, both sites are impacted simultaneously with cold spells. Long cold spells (30 days) are observed in 2009 and 2018, leading to an anomaly of more than -1.9°C.

The cold spell observed in 2018 in the Bay of Vilaine may explain the later IPGP. There is no change in the IPGP in 2011 and 2013, despite the cold spell, the period of occurrence being too early during winter 2011, and the duration too short in 2013 (only 10 days).

In the Bay of Brest, the cold spells in 2003 and 2004 may explain the delay of the IPGP (respectively days 93 and 85). The presence of long and intense cold spells in 2010 and 2011 do not shift the IPGP (days 50 and 67) because they occur too early (before day 20).

**3.4.2 Wind bursts**

Based on our model simulations, the wind bursts that occur during at least three continuous days have no impact on the IPGP in both bays, whatever the duration, the period and the intensity (+/- 1 day). In the Bay of Vilaine, only one wind event is detected in 2018 (3 days long and 6 m.s$^{-1}$). In the Bay of Brest, several events are detected, but no significant impact is observed on the IPGP.

**3.4.3 Flood events**

River floods can delay the IPGP by resuspending sediment in the water column and therefore limiting light penetration in the water column. Inputs of nutrients have no impact during the late winter period because nutrient concentrations are maximal, with no limitation on phytoplankton growth. Flood events are analyzed with observation data collected in the month prior to the IPGP date because the 1DV modeling approach does not allow the sensitivity to hydrological events to be simulated (*i.e.* it is necessary to simulate horizontal advection processes).

In the Bay of Brest, the impact of flood events depends on their duration and intensity: when the flood exceeds 15 days, a delay in the IPGP is detected. Shorter and more intense floods (> 300 m$^3$ s$^{-1}$) do not impact the IPGP.

In the Bay of Vilaine, only two flood events are observed close to the IPGP date in 2014 and 2015. The 2015 flood event, which is 10 days longer and more intense (> 100 m$^3$ s$^{-1}$) than the 2014 one, delays the IPGP date by 10 days.



**4 Discussion**

4.1 Comparison of the phytoplankton growing period in both bays

Despite their contrasting hydrodynamics (*e.g.* Petton *et al.*, 2020; Poppeschi *et al.*, 2021; Lazure and Jegou, 1998; Ratmaya *et al.*, 2019; Menesguen *et al.*, 2019), the median dates of the start and the end of the productive period are the same in the bay of Brest and in the bay of Vilaine whether they are calculated from high-
and low- frequency datasets and from model simulations. The phytoplankton growing period occurs from March to September and lasts about 190 days in both bays. This concordance is related to a similar seasonality of the environmental drivers.

The observed cumulative fluorescence is almost double in the Bay of Vilaine compared with the Bay of Brest. This difference in the amount of chlorophyll produced in surface waters from both bays is also recorded by
the low-frequency observation programs and by satellite observations (Menesguen *et al.*, 2019). It can be explained by the difference of the hydrodynamics and the influence of different watersheds. The Bay of Brest is a semi-enclosed bay with a macro-tidal regime influenced by two local rivers (Aulne and Elorn) whereas the Bay of Vilaine has a weaker tidal regime, is open on the continental shelf and is widely influenced by a large river (Loire river).

Two different patterns of the phytoplankton growing period are identified by the k-means classification in both bays. The flattened, weak and long bloom highlighted in the Bay of Brest can be explained by assuming that nutrients are not limiting the phytoplankton growth during spring. The maintenance of the diatom succession throughout spring since the 1980's (Quéguiner 1982, Del Amo et al 1997) can be explained by the combination of increasing N and P loads, intense Si recycling and a macrotidal regime (Ragueneau *et al.*, 2019). The
phytoplankton growing period in the bay of Vilaine is characterized by several successive peaks including two main ones. Nutrients here drive the seasonal evolution of the phytoplankton growing period through periods of nutrient-limited conditions. These fluctuations are governed by phosphorus and nitrate loads from Vilaine and Loire rivers (Ratmaya *et al.*, 2019), but probably also by the stoichiometry of recycled elements in the water and at the water-sediment interface (Ratmaya *et al.*, 2022).


4.2 Identification of the environmental conditions supporting the IPGP

The method that we developed to detect IPGP on both high-frequency and low-frequency *in situ*
observations shows comparable results and detects similar initiation dates for some years, while a time lag between high- and low-frequency observations can be observed for other years. This difference is mainly explained by the difference in the sampling frequency. The late deployment of the buoy in the Bay of Vilaine (i.e. not deployed until mid-February before 2018) can also explain some differences between both sites. High-frequency data provide a more accurate detection of the day of the IPGP, while an uncertainty of about $\pm$ 7 days is observed with
low-frequency observations. This comparison between high- and low-frequency based IPGP detection highlights the sensitivity of sampling strategy in the observation of phytoplankton growing periods (Bouman *et al.*, 2005; Serre-Fredj *et al.*, 2021) related to the response of the ecosystem within a few hours after an environmental change (Lefort and Gasol, 2014; Thyssen *et al.*, 2008).

The modeled IPGP, based on the year 2015, is coherent with high-frequency observations (around 5 days of difference between modeled and observed IPGP). Considering the idealized framework for modeling computations (1DV model instead of a realistic 3D model configuration), the agreement between observations and simulations validates the 1DV approach to explore IPGP dynamics. With the 1DV configuration, the vertical dynamics in the water column, coupled with biogeochemistry and sediment dynamics are well reproduced.
Atmospheric forcings and interactions with the bottom layer are the main environmental drivers. The full range of impacts related to the horizontal advection (*e.g.* in considered regions, rivers advected plumes can change the hydrodynamics and the biogeochemical contents) are not evaluated, however. In the Bay of Brest and in the Bay of Vilaine, such advected sources exist (*e.g.* Poppeschi *et al.*, 2021; Lazure and Jegou, 1998) but inputs from rivers are not main drivers of the IPGP in nutrient-rich environments. Nutrient loads advected by rivers may impact the
phytoplankton community during the growing period rather than at IPGP (*e.g.* Ratmaya *et al.*, 2019).

We characterize similar environmental conditions in both bays as the IPGP is mainly driven and limited by similar large-scale conditions. The ideal temperature (> 10°C) and PAR (1300 W m$^{-2}$) for the IPGP are in agreement with those from previous studies conducted in similar coastal ecosystems (*e.g.* Glé *et al.*, 2007;
Townsend *et al.*, 1994; Trombetta *et al.*, 2019). Neap tidal conditions (tidal amplitude about 4 m), weak wind (lower than 3 m s$^{-1}$) and weak river flow can also play a positive role to observe earlier IPGP according to the previous study of Ragueneau *et al.*, 1996.





As also shown in the German Bight (Tian *et al.*, 2011), wind intensity is a driver of turbidity in the water column which inhibits phytoplankton growth. The impact of wind direction on the IPGP is estimated to be negligible.

Local changes in those features (temperature, incident radiation, tidal conditions, wind conditions and river flow) induce differences in detected IPGP.

The comparison of the individual importance of each environmental driver shows that temperature is the key environmental driver in both bays. Similarly in the North Sea, Wiltshire *et al.* (2015) highlight the importance

of the light availability in the timing and intensity of the spring bloom. However, too high turbidity (due to sediment resuspension) can also limit the production and delay IPGP in the bay of Vilaine. Similar limitations are observed in the German Bight (Tian *et al.*, 2009) or along the UK South Coast (Iriarte and Purdie, 2004). The combined effect of surface incident radiation and turbidity can amplify the delay of the IPGP. However, with the existence of minimum mandatory conditions, an earlier IPGP can not be observed or modeled, except if thresholds

are reached earlier (e.g. warmer temperature earlier during the year).

4.3 Interannual evolutions of the IPGP

The IPGP in these two bays shows a strong interannual variability with initiation dates varying from late

winter to spring. A mean difference of 50 days between the earliest and latest IPGP dates is observed. Each year has a different date of IPGP related to different environmental conditions. However, the beginning of the phytoplankton growing period is always dominated in both bays by the same centric diatoms, genera *Chaetoceros* and *Skeletonema*, whose abundance varies from year to year depending on climatic conditions (REPHY, 2021)

The earliest IPGP are observed when the environmental conditions are favorable early in the year. For

example, the IPGP occurs before day 50, both in 2010 in the Bay of Brest and in 2012 in the Bay of Vilaine, associated with exceptionally weak wind and river flow in addition to a sufficient PAR and nearly-optimal temperature of around 10°C. But if the environmental conditions are not favorable, such as in 2017 and 2019 in both bays, the IPGP is delayed. This can be due to a strong wind during several days (not a single wind burst) and a weak PAR and sometimes also because of turbidity events.

The reason for the IPGP advance or delay is not always the same. For example in 2003 in the Bay of Brest, the IPGP was late due to low temperature conditions, rather than a strong wind or a lack of PAR as seen previously. The IPGP can be different from one bay to another in the same year, almost half of the years studied (2012, 2013, 2014, 2015, 2016). For example, the 2012 IPGP is early in the Bay of Vilaine (day 53 while it is later in the Bay of Brest (day 80), related to strong wind activity and low PAR. This difference between the two bays

indicates a local, not regional, effect of the processes affecting the IPGP.

The analysis of the IPGP over the last two decades has highlighted its evolution through two trends, one per decade. The IPGP occurs earlier each year until 2010 when the trend is reversed. At a larger scale, this change in trends is not directly observed for the same years. For example, using hourly data, Hunter-Cervera *et al.* (2016)

show earlier blooms of picophytoplankton on the New England Shelf during 2003-2012 due to warming spring periods, and later blooms in 2013-2015 for cooler spring temperatures. The similarity between these observations and those found here in our study on the other side of the Atlantic basin for a slightly later breaking year (2012 instead of 2010) suggests a large-scale impact of the warming waters in spring. On the eastern part of the Atlantic, we also know that 2010 was an atypical year, with an important accumulation of phytoplankton biomass as

observed by Bedford *et al.* (2020) on the North-West European shelf. However, limited indicators do not allow conclusions to be drawn regarding the impact of large-scale forcings on observed shifts in phytoplankton blooms.

As the climate warms, earlier phytoplankton blooms are expected (Friedland *et al.*, 2018) but not later IPGP as observed in our study regions. However, the mechanisms that trigger blooms in coastal ecosystems -

especially eutrophic ones - are not similar to the processes that influence blooms in the open ocean. For example, by investigating long-term (1975-2005) daily data, Wiltshire *et al.* (2008) observe later phytoplankton blooms in the German bight, but with no link to global warming. Henson *et al.* (2018) model a bloom shift of 5 days per decade from 2006 to 2025, with later blooms. A possible explanation may involve the lower spring sea surface temperatures, as observed in recent years (Hunter-Cervera *et al.* 2016), which could cause a delay of the IPGP.

We do not detect significant trends in environmental conditions over the last 20 years at either site, and therefore do not establish direct links with the trends observed in the IPGP timing. In the southern California Bight, similar changes in IPGP are observed from 1983 to 2000, but no link with environmental drivers has been identified (Kim *et al.*, 2009).

4.4 Extreme events



We show that a cold spell is likely to delay the IPGP if it occurs at the end of winter (after 20th February) or/and if the cold spell lasts long enough (> 15 days). This is in accordance with the study of Gomez and Souissi (2008) in the English Channel where cold spells can affect the date of IPGP by increasing the water column mixing.

In both bays, the drop in temperature related to the cold spell prevents the IPGP. Cold spells may also drive local patterns by influencing the phytoplankton communities (Gomez and Souissi, 2008; Schlegel *et al.*, 2021).

Flood events have an influence on the phytoplankton biomass when they occur in spring due to the supply of nutrients. When they occur in late winter, nutrients are already at their maximum. The impact of floods on IPGP is consequent only if they are at least 15 days long. This scheme is also observed by Saeck *et al.* (2013) along a

river-estuary-bay continuum and explained by a shortened water residence time and a limited light due to flood-induced turbidity in the coastal zone.

No relationship is observed between wind events and IPGP in both bays because they are weakly stratified contrary to open seas (*i.e.* Black Sea, Mikaelyan *et al.*, 2017). In coastal stratified regions (e.g. under the influence of river plumes), strong wind and tidal mixing can enhance the mixing and break down stratification. Such

conditions can also enhance phytoplankton production (Joordens *et al.*, 2001). During the IPGP, except during floods, both regions are weakly stratified and are then less sensitive to combined wind/tidal short events.

## 5 Conclusions

This study provides a new understanding of the IPGP in coastal areas. Our results allow us to characterize

the IPGP in two different eutrophic bays on the basis of both high and low-frequency *in situ* data, in combination with simulations from a 1DV model. Strong similarities are found in both bays. An important interannual variability of the IPGP is observed, with a trend towards a later IPGP over the last decade (2010-2020). We quantify the importance of environmental conditions on the IPGP, with water temperature and turbidity being the main drivers over wind intensity and surface incident radiation. The IPGP is a complex mechanism, usually

triggered by more than one environmental parameter. The analysis of the influence of extreme events reveals that cold spells and floods have a strong impact by delaying the IPGP when episodes are long enough and occur after winter. No effect of wind bursts is detected.

While our analysis shows comparable IPGP dynamics when based on either simulations from a 1DV model or *in situ* observations, we will next investigate the effect of horizontal advection on phytoplankton

dynamics using a 3D realistic model. We will focus on the exploration of the variability of phytoplankton communities during the IPGP in order to evaluate whether a community shift occurs, as observed in other studies and for other ecosystems (Ianson *et al.,* 2001; Edwards and Richardson, 2004; Chivers *et al.,* 2020). The lack of long-term time series of zooplankton is a clear hindrance when investigating the top-down control on the IPGP. The investigation of other contrasted coastal environments will allow us to better understand and anticipate the

expected impact of global change on coastal phytoplankton dynamics.

**Author contributions**

CP, GC, AD, RV, PR-M and EGo conceptualized the study. PR-M, EGr and MR collected data. MP and GC developed the model configuration. CP, GC, AD and RV drafted the first versions of the paper. CP carried out all the analyses and wrote the final version of the paper. All authors contributed to the discussions and revisions of

the study.

**Acknowledgements**

We would like to acknowledge COAST-HF (http://www.coast-hf.fr), SOMLIT (http://somlit.epoc.u-bordeaux1.fr) and REPHY (https://doi.org/10.17882/47248) national observing networks, for providing data flux readily available. COAST-HF and SOMLIT are components of the National Research Infrastructure ILICO. We

would like to thank the Shom for tidal data and also Météo-France for wind and solar flux products. We also thank Dr Sally Close for her proofreading.

**Financial support**

This study is part of the State-Region Plan Contract ROEC supported in part by the European Regional Development Funds and the COXTCLIM project funded by the Loire-Brittany Water Agency, the Brittany region

and Ifremer.



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












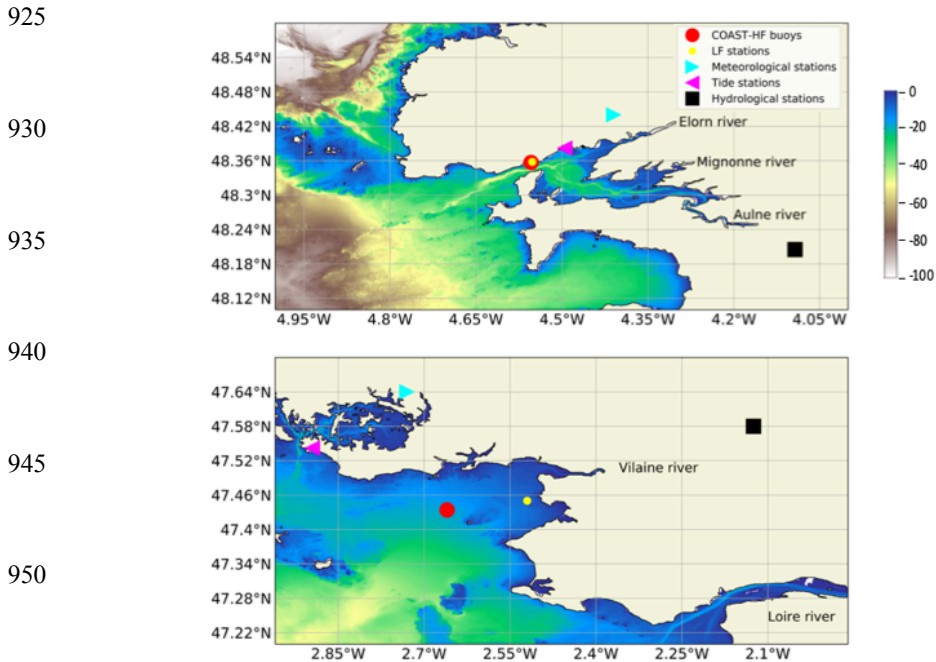

**Figure 1: Location of the sampling sites: COAST-HF-Iroise and COAST-HF-Molit buoys (red circles); SOMLIT-Brest and REPHY-Loscolo sampling stations (yellow circles); Brest and Crouesty tide gauge stations (blue triangles); Guipavas and Vannes-Séné meteorological stations (purple triangles); hydrological stations of the Aulne and Vilaine rivers (black squares) with the Loire station off the map.**


| Parameters | Bay of Brest | Bay of Vilaine |
|---|---|---|
| Dissolved $O_2$ *(mg $L^{-1}$)* | 9 | 10 |
| Mesozooplankton *($\mu molN\ L^{-1}$)* | 0.05 | 0.1 |
| Microzooplankton *($\mu molN\ L^{-1}$)* | 0.05 | 0.05 |
| Dinoflagellates *($\mu molN\ L^{-1}$)* | 0.05 | 0 |
| Diatoms *($\mu molN\ L^{-1}$)* | 0.5 | 0.5 |
| Soluble reactive phosphorus *($\mu mol\ L^{-1}$)* | 0.5 | 0.8 |
| Silicic acid *($\mu mol\ L^{-1}$)* | 10 | 30 |
| Nitrate *($\mu mol\ L^{-1}$)* | 16 | 30 |
| Ammonium *($\mu mol\ L^{-1}$)* | 0.5 | 0.25 |
| Coarse sand *(g $L^{-1}$)* | 0 | 0 |
| Fine sand *(g $L^{-1}$)* | 0 | 0 |





| Mud (g L⁻¹) | 0.03 | 0.05 |
| --- | --- | --- |

**Table 1: Initial conditions in the water column for the MARS-1DV model for the beginning of the simulation on the February 15ᵗʰ.**


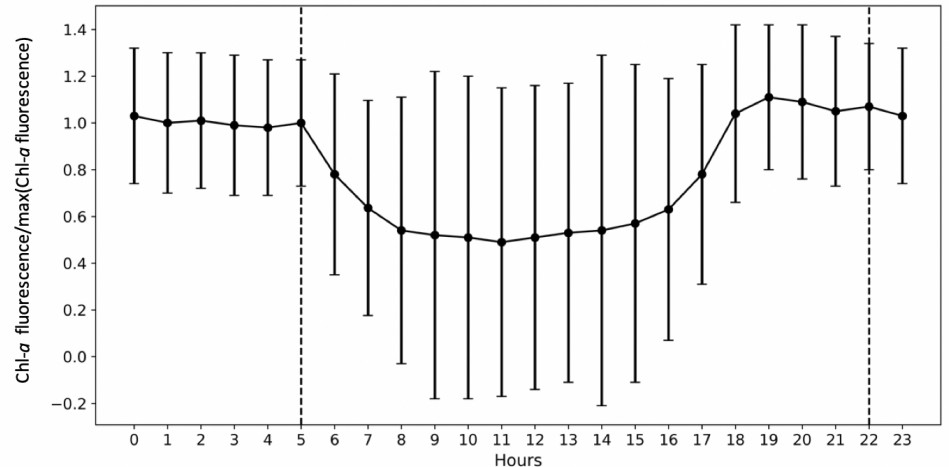

**Figure 2: Importance of the Quenching effect on Chl-*a* fluorescence is represented by COAST-HF-Iroise data from 2000 to 2019. The standard deviation is represented by vertical black bars. The dashed lines represent the beginning and end of the selected values for the rest of the study from 10 pm to 5 am.**


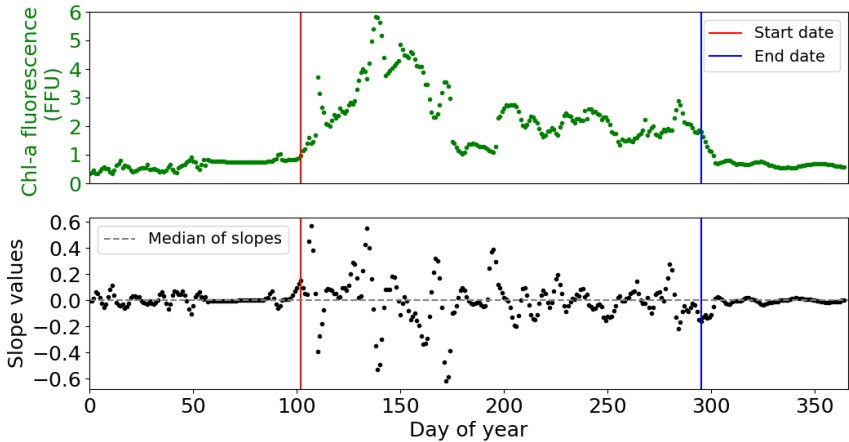

**Figure 3: Example of detection of the start (red line) and end (blue line) of the phytoplankton growing period in 2001 at COAST-HF-Iroise. The threshold value - median of slopes - is represented by a dotted grey line.**




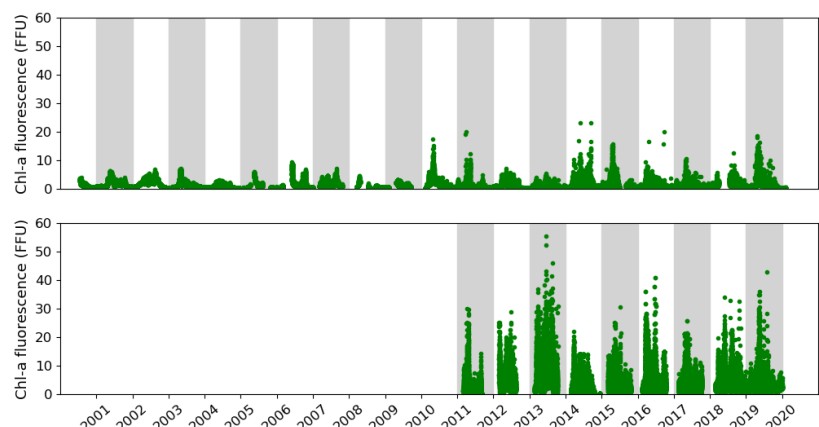

**Figure 4: Temporal changes in the *in situ* Chl-*a* fluorescence measured in the Bay of Brest (top) and the Bay of Vilaine (bottom).**


| | Start date (Day of year) | End date (Day of year) | Duration (Days) | Cumulative Chl-*a* fluorescence (FFU) |
|---|---|---|---|---|
| | *Min - **Median** - Max* | *Min - **Median** - Max* | *Min - **Median** - Max* | *Min - **Median** - Max* |
| **Bay of Brest** (2001-2019) | 50 - **69** - 102 | 253 - **274** - 308 | 165 - **200** - 256 | 217 - **364** - 567 |
| **Bay of Vilaine** (2011-2019) | 53 - **68** - 93 | 218 - **269** - 316 | 165 - **179** - 239 | 276 - **582** - 1406 |

**Table 2: Global characteristics of the phytoplankton growing period in the Bay of Brest and in the Bay of Vilaine.**


**(a)**

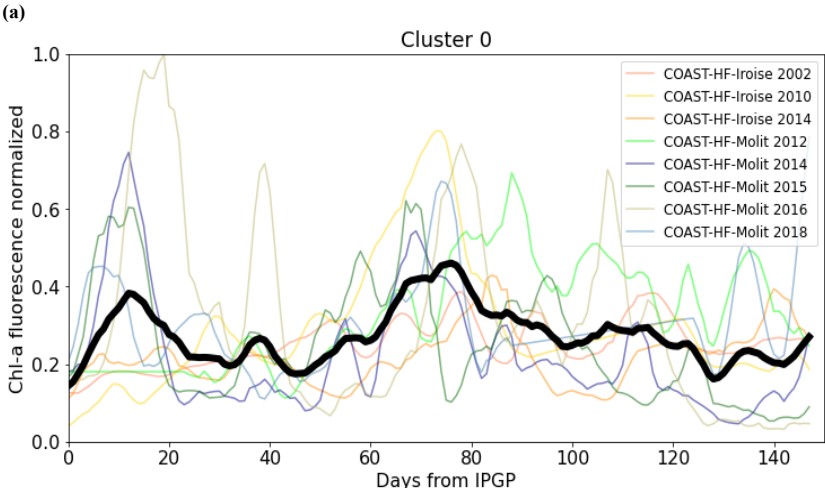





**(b)**

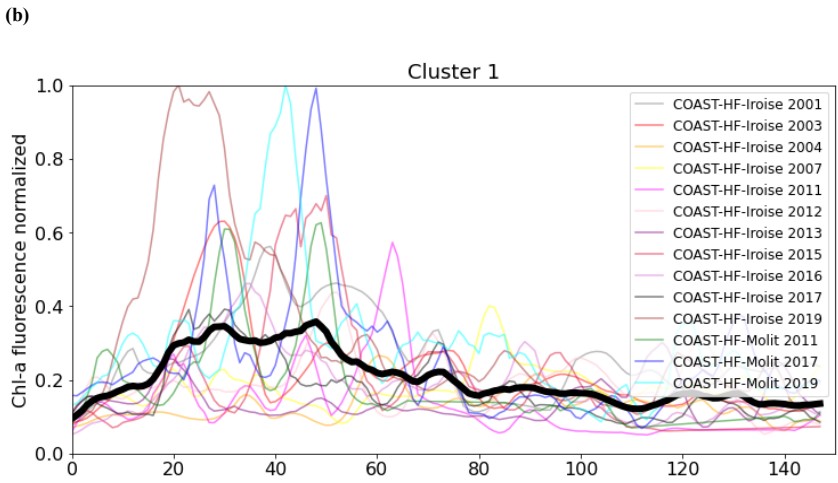

**Figure 5: (a) Cluster 0 and (b) cluster 1 representative of the patterns of the phytoplankton growing period observed in both bays. The median pattern is drawn in bold.**


| Year | 2001 | 2002 | 2003 | 2004 | 2005 | 2006 | 2007 | 2008 | 2009 | 2010 | 2011 | 2012 | 2013 | 2014 | 2015 | 2016 | 2017 | 2018 | 2019 |
|---|---|---|---|---|---|---|---|---|---|---|---|---|---|---|---|---|---|---|---|
| **Bay of Brest** COAST-HF-Iroise | 1 | 0 | 1 | 1 | | | 1 | | | 0 | 1 | 1 | 1 | 0 | 1 | 1 | 1 | | 1 |
| **Bay of Vilaine** COAST-HF-Molit | | | | | | | | | | | 1 | 0 | X | 0 | 0 | 0 | 1 | 0 | 1 |

**Table 3: Cluster group assigned to each annual phytoplankton growing period on both sites. Grey boxes represent years with missing data. The cross represents the year 2013 of the Bay of Vilaine not considered.**


**(a)**

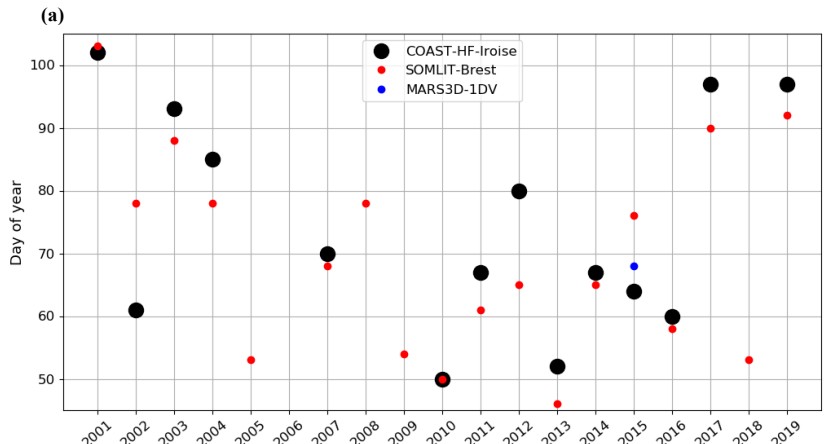


**(b)**



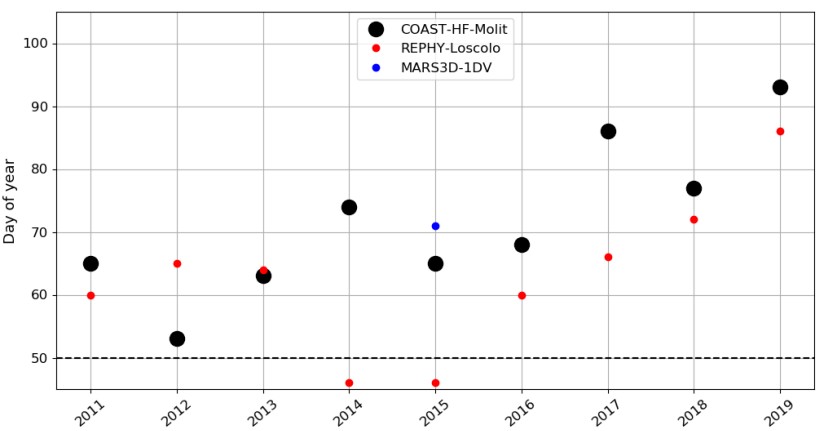

**Figure 6: Changes in the IPGP date in (a) the Bay of Brest and (b) the Bay of Vilaine are determined with high-frequency time series (black circles), low-frequency time series (red circles) and with the model (blue circle). The dotted black line represents the date of the COAST-HF-Molit buoy deployment.**













**(a)**

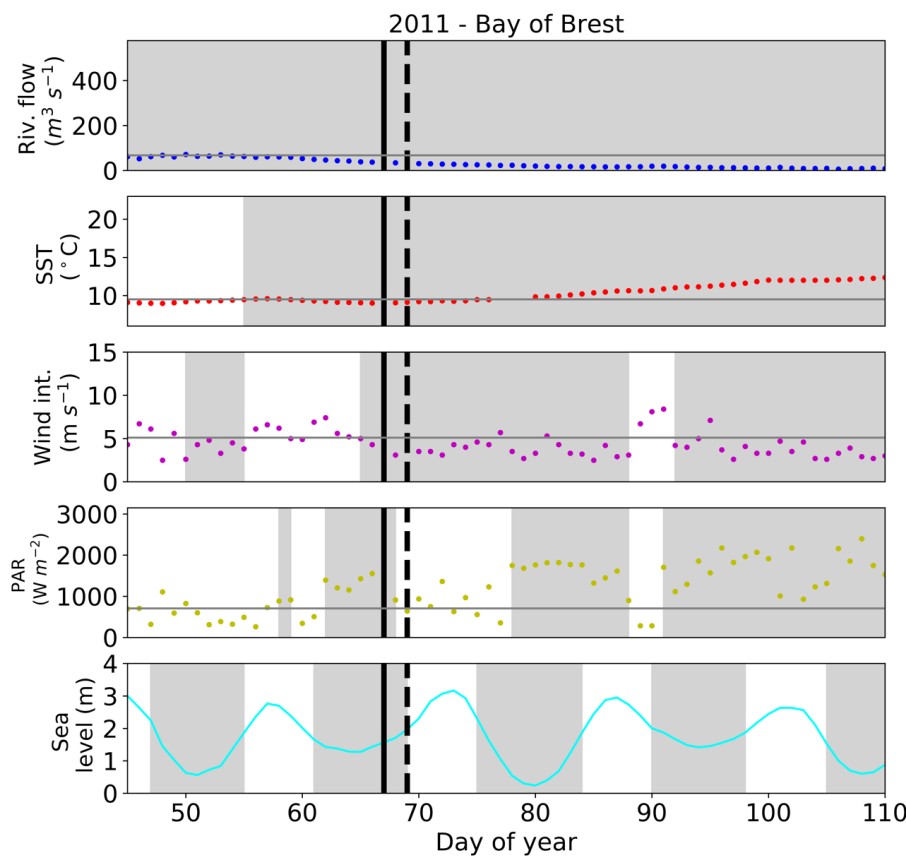





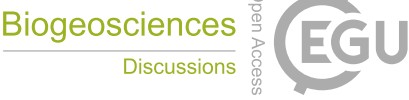

**(b)**

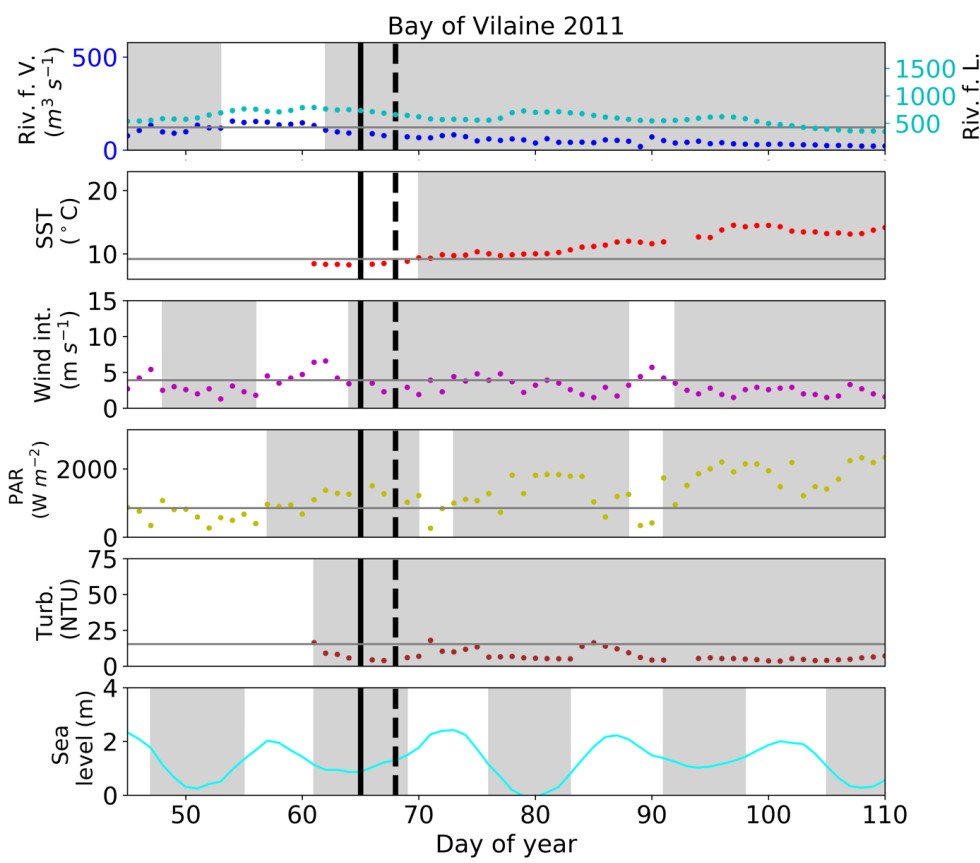







**(c)**

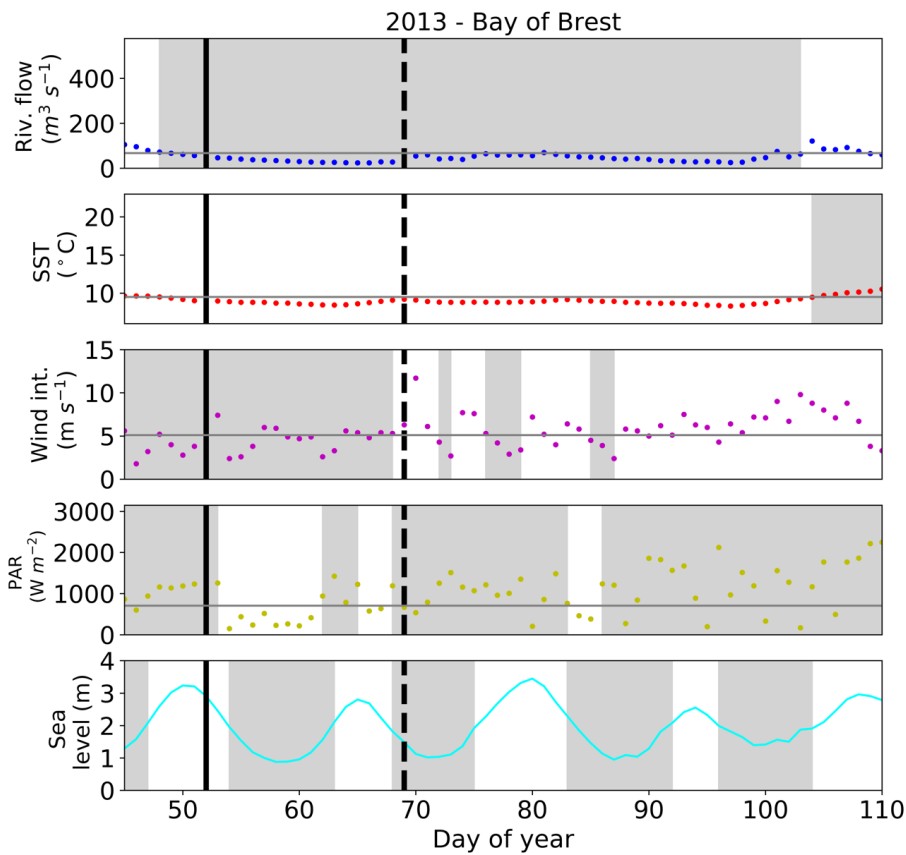








**(d)**

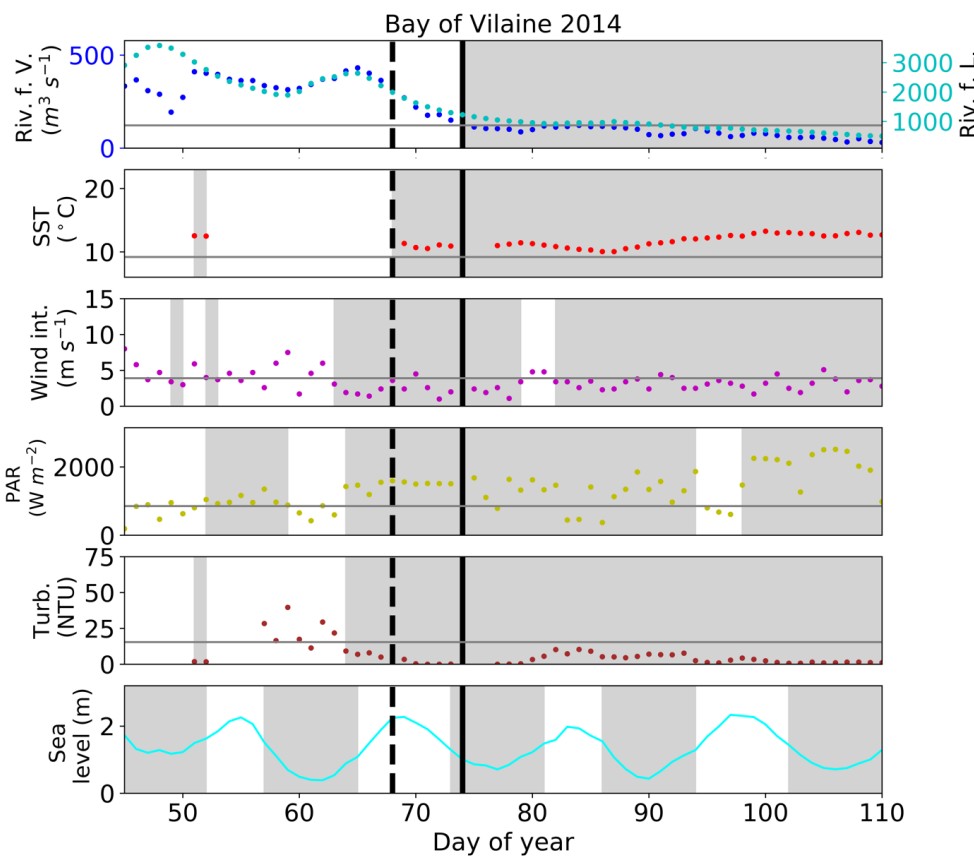

Figure 7: Examples of change in the Chl-a fluorescence and environmental drivers: flow of the Aulne, Vilaine and Loire rivers, Sea Surface Temperature (SST), wind intensity, PAR, turbidity and sea level. Year 2011 is characterized by a mean IPGP date in (a) the Bay of Brest and (b) the Bay of Vilaine; Year 2013 by an early IPGP date in (c) the Bay of Brest; Year 2014 by a late IPGP date in (d) the Bay of Vilaine. The mean IPGP date of each bay is represented by a dotted black line and the IPGP date of the year is represented by a straight black line. Thresholds of each environmental driver are represented by grey vertical lines corresponding to the mean conditions calculated $\pm$ 15 days around the IPGP date. Grey areas are time periods favorable to IPGP.

| | River flow (m³ s⁻¹) | SST (°C) | Wind intensity (m s⁻¹) | PAR (W m⁻²) | Turbidity (NTU) | Sea level (m) |
|---|---|---|---|---|---|---|
| | Min - **Median** - Max | Min - **Median** - Max | Min - **Median** - Max | Min - **Median** - Max | Min - **Median** - Max | Min - **Median** - Max |
| **Bay of Brest** (2001-2019) | 13 - **46** - 100 | 8 - **10** - 12 | 1 - **3** - 6 | 915 - **1373** - 2220 | 1 - **7** - 21 | 0.5 - **1.6** - 2.9 |
| **Bay of Vilaine** (2011-2019) | 36 - **96** - 205 | 8 - **10** - 11 | 1 - **3** - 4 | 814 - **1341** - 1939 | 0 - **7** - 22 | 0.6 − **0.9** - 1.6 |

Table 4: Characteristics of environmental drivers at the date of IPGP in the Bay of Brest and in the Bay of Vilaine.





**(a)**

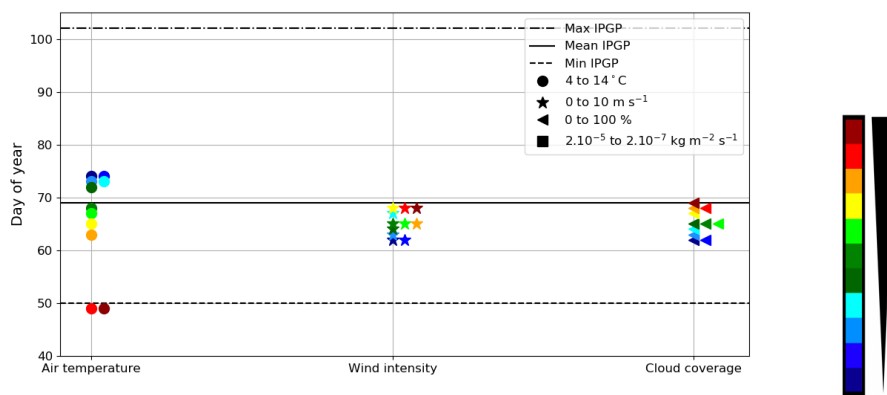

**(b)**

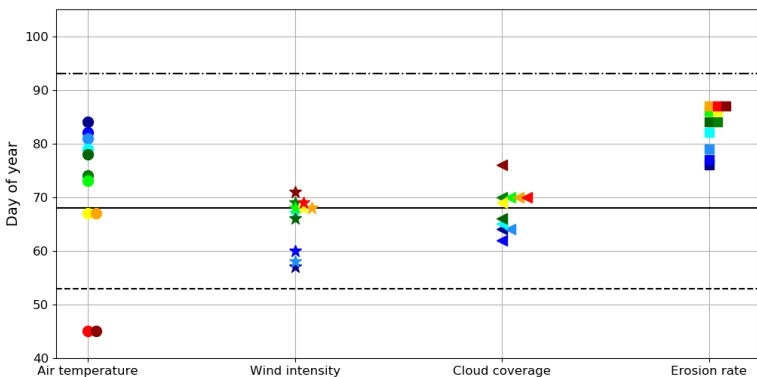


**Figure 8: Impact of the variation of environmental drivers on the date of IPGP in (a) the Bay of Brest and (b) the Bay of Vilaine. Steps of: 1°C for the air temperature, 1 m s⁻¹ for the wind intensity, 10 % for the cloud coverage and 0.0000036 kg m⁻² s⁻¹ for the erosion rate equivalent to a variation of suspended matter between 0.02 and 0.08 mg L⁻¹ at**
**IPGP.**





| Experiment | Air temperature (°C) | Wind intensity (m s⁻¹) | Cloud coverage (%) | Erosion rate (kg m⁻² s⁻¹) | Simulated IPGP Bay of Brest (days) | Simulated IPGP Bay of Vilaine (days) |
|---|---|---|---|---|---|---|
| 1 | 4 | 3 | 70 | $2.10^{-6}$ | +5 | +16 |
| 2 | 14 | 3 | 70 | $2.10^{-6}$ | -20 | -23 |
| 3 | 10 | 0 | 70 | $2.10^{-6}$ | -1 | -11 |
| 4 | 10 | 10 | 70 | $2.10^{-6}$ | -7 | +3 |
| 5 | 10 | 3 | 0 | $2.10^{-6}$ | = | -4 |
| 6 | 10 | 3 | 100 | $2.10^{-6}$ | -7 | +8 |
| 7 | 10 | 3 | 70 | $2.10^{-7}$ | | +8 |
| 8 | 10 | 3 | 70 | $2.10^{-5}$ | | +19 |
| A | 4 | 10 | 100 | $2.10^{-5}$ | +9 | +64 |
| B | 4 | 10 | 70 | $2.10^{-6}$ | +5 | +17 |
| C | 4 | 3 | 100 | $2.10^{-6}$ | +5 | +28 |
| D | 10 | 10 | 100 | $2.10^{-6}$ | = | +6 |
| E | 4 | 10 | 70 | $2.10^{-5}$ | | +48 |
| F | 4 | 3 | 100 | $2.10^{-5}$ | | +46 |
| G | 10 | 10 | 100 | $2.10^{-5}$ | | +34 |
| H | 10 | 3 | 100 | $2.10^{-5}$ | | +19 |
| I | 10 | 10 | 70 | $2.10^{-5}$ | | +29 |
| J | 4 | 3 | 70 | $2.10^{-5}$ | | +36 |
| K | 14 | 0 | 0 | $2.10^{-7}$ | -20 | -11 |
| L | 14 | 0 | 70 | $2.10^{-7}$ | -21 | -11 |
| M | 14 | 3 | 0 | $2.10^{-7}$ | -20 | -11 |
| N | 10 | 0 | 0 | $2.10^{-7}$ | -11 | -11 |

Table 5: Assumptions are explored in the 1DV model for environmental parameters independently (1-8) and with combined effect (A-N) with the modified values (grey background) and text in bold for the Bay of Brest only (+ for later IPGP, - for earlier IPGP, = for equal IPGP) with IPGP equal the mean observed IPGP of day 68.

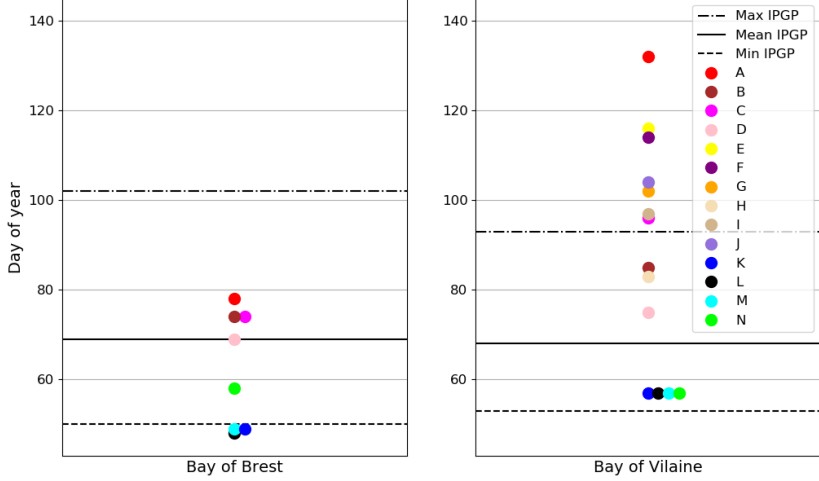


Figure 9: Influence of combined environmental parameters for the MARS-1DV model in both bays (Bay of Brest - left and Bay of Vilaine - right) with detailed experiments in Table 2.

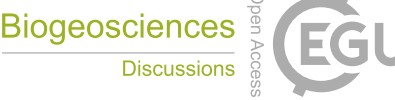

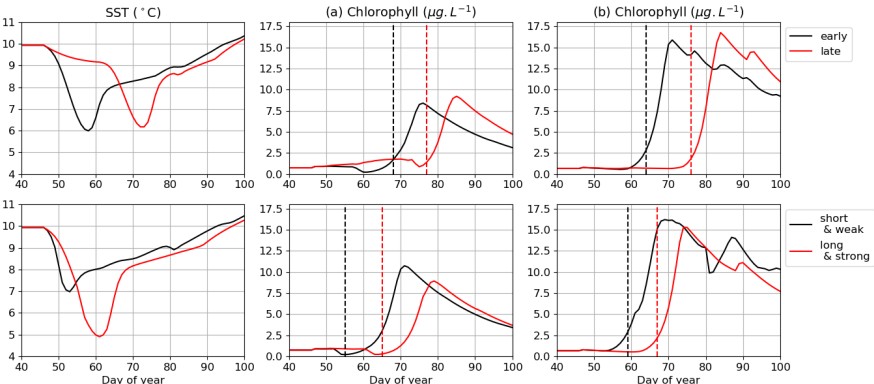

**Figure 10: Impact of cold spells on the IPGP date simulated in (a) the Bay of Brest and (b) the Bay of Vilaine. Four conditions of cold spells are explored: an early (mid-February), a late (end of February), a short (8 days) and a long (20 days). The IPGP dates are represented by dotted lines.**