# Peer review of "Interannual variability of the initiation of the phytoplankton 1"

_Biogeosciences, 2022_

## Author Comment (AC1)

Biogeosciences Discuss., referee comment RC1 https://doi.org/10.5194/bg-2022-86-RC1, 2022 © Author(s) 2022. This work is distributed under the Creative Commons Attribution 4.0 License.

**Comment on bg-2022-86**

Anonymous Referee #1

Referee comment on "Interannual variability of the initiation of the phytoplankton growing period in two French coastal ecosystems" by Coline Poppeschi et al., Biogeosciences Discuss., https://doi.org/10.5194/bg-2022-86-RC1, 2022

*Main manuscript modifications are highlighted in red.*

In this manuscript the decadal years variability of the IPGP (Initiation of the Phytoplankton Growing Period) is analyzed at two coastal stations located in the northern (Iroise Sea) and eastern Bay of Biscay (Bay of Vilaine). The phytoplankton biomass is related to fluorescence measured by instrumented buoys. The sensitivity analysis for identifying the major causes of the variability is made through a 1-D biogeochemical model applied to the year 2015. The major result of this work is that, despite different environments, the IBGP days are very similar at the two stations. No significant trend in the IPGP is observed in the time series However the variability is high and seems to show an earlier IGBP in the middle of the period -around year 2010- (at about day 60 against day 90 for the beginning and the end of the period). The results are interesting but this manuscript presents some flaws.

We thank the reviewer for the constructive reviews. We considered each point below.

The data used are fluorescence-derived Chlorophyll-a, the identification of the factors influencing the IPGP is made through a model and the conclusions are evasive.

The conclusions have been rephrased to be more detailed and less evasive in the first paragraph of part 5 conclusions and in the abstract. We then describe main conclusions built on model results but also on *in situ* observations.

On the first point, auxiliary analysed Chl-a concentrations collected bimonthly (which is a large interval for this purpose) corroborate nevertheless the IBGP derived from fluorescence data. The main issue comes therefore from the discussion based on model results. The model is considered as perfect and the causes of the variability are discussed from its outputs. From the introduction to the conclusion and throughout the discussion the real issues of the IGBP have not been considered with sufficient care. Blooms at local stations in river plumes may or may not occur, the true question is what happens at large scale? What is the connection with the dynamics of phytoplankton in the whole area?

The aim of this study was to investigate the initiation of the IPGP in nearshore waters under the influence of rivers highly rich in nutrients. The bathymetry (< 30 m) and the hydrodynamics of the Bay of Vilaine (Mor Braz) and the Bay of Brest do not allow them to be compared at larger scale respectively to the Bay of Biscay and to the Iroise sea. The late winter phytoplankton blooms reported in the northern Bay of Biscay by Labry et al.

(2001) and Gohin et al. (2003) were observed respectively in isobaths 60-30 and 120-80 m. According to the satellite observations, the phytoplankton dynamic in nearshore of south Brittany (Gohin 2010; Gohin 2012; marc.ifremer.fr website) is clearly different from the rest of the Bay of Biscay while the difference between the Gironde plume and the Bay of Biscay is less true (Figure 1). The maps of the monthly mean chl-a concentrations, as well as the annual cycles of chl-a, also show that it is in the nearshore of the Bay of Biscay, and in particular in the plumes of the Loire and Vilaine rivers, that the concentrations are the highest. For all these reasons, it does not seem appropriate to extend spatially (at the scale of the continental shelf of the bay of Biscay) our results obtained at one nearshore HF instrumented station.

In the manuscript introduction with the objectives have been modified to explain the local scale addressed in the study. We also add a specific sentence in section 2.1 "coastal temperate ecosystems".

[Figure]

*Figure 1: Mean Chl-a concentrations in January/February during 2003-2010 (from Gohin, 2012)*

*Gohin Francis, Saulquin Bertrand, Bryere Philippe (2010). Atlas de la Température, de la concentration en Chlorophylle et de la Turbidité de surface du plateau continental français et de ses abords de l'Ouest européen.https://archimer.ifremer.fr/doc/00057/16840/*

*Gohin Francis (2012). Répartition spatio-temporelle de la chlorophylle a. Sous-région marine Golfe de Gascogne. Evaluation initiale DCSMM. MEDDE, AAMP, Ifremer , Ref. DCSMM/EI/EE/GDG/12/2012, 13p. https://archimer.ifremer.fr/doc/00329/44009/*

*Labry Claire, Herbland Alain, Delmas Daniel, Laborde P, Lazure Pascal, Froidefond J, Jegou Anne-Marie, Sautour B (2001). Initiation of winter phytoplankton blooms within the Gironde plume waters in the Bay of Biscay. Marine Ecology Progress Series, 212, 117-130.*

*Morin, P., Le Corre, P., Marty, Y., L'Helguen, S., 1991, Evolution printanière des éléments nutritifs et du phytoplancton sur le plateau continental armoricain (Europe du Nord-Ouest). Oceanologica Acta 14:263-279.*

We know since the end of the 90's that strong early blooms within the Gironde plume may consume a large part of the winter Phosphorus stock at the beginning of March or even earlier. This has been attested also by satellite observations at broader scale.

According to the results of Labry et al. (2001) - Table 1, we agree that the late winter bloom consumes a large part of the winter phosphorus stock at the beginning of March within the Gironde plume. The situation is completely different in the Bay of Vilaine and in the Bay of Brest where nutrient concentrations remain high until late March. When Labry et al. (2001) observed phosphate concentrations close to the limit of detection of the analytical method (< 0.05 µmol/l) in February, the median phosphate concentration is equal to 0.83 µmol/L in the Bay of Vilaine and to 0.43 µmol/L in the Bay of Brest (Figure 2). The median silicate concentration (Figure 2, respectively 38.1 and 8.1 µmol/L in the Bay of Vilaine and the Bay of Brest) is highly above the half-saturation constant required for their assimilation by diatoms (Ks = 2 µmol/L, Del Amo and Brzezinski 1999). The waters are also not limited by silicate before the IGPG date, which accredits the presence of large diatoms at the IPGP date and not a phytoplankton population dominated by small cells.

| | $PO_4$ µM | $NO_3$ µM | $NH_4$ µM | $Si(OH)_4$ µM | $NO_3:PO_4$ at:at | $N_{min}:PO_4$ at:at | $NO_3:Si$ at:at | $n$ |
|---|---|---|---|---|---|---|---|---|
| **Turbid plume** | | | | | | | | |
| January | 0,586 ± 0,086 | 26,2 ± 3,7 | 1,7 ± 1,5 | 17,9 ± 3,3 | 47 ± 6 | 50 ± 8 | 1,4 ± 0,1 | 4 - 5 |
| Start March | 0,215 ± 0,064 | 13,1 ± 1,5 | 0,2 ± 0,2 | 8,1 ± 2,2 | 67 ± 25 | 69 ± 26 | 1,7 ± 0,6 | 4 |
| **« girondines » waters** | | | | | | | | |
| January | 0,320 ± 0,087 | 9,9 ± 3,6 | 1,1 ± 1,0 | 7,5 ± 2,0 | 31 ± 9 | 36 ± 9 | 1,3 ± 0,2 | 17 - 19 |
| Start March | < 0,020 | 4,9 ± 2,6 | 0,1 ± 0,2 | 1,9 ± 1,0 | > 100 | > 100 | 2,4 ± 1,1 | 20 - 22 |
| June | < 0,020 | 0,8 ± 0,5 | 0,3 ± 0,2 | 2,3 ± 0,6 | - | - | 0,4 ± 0,4 | 21 - 23 |
| **« oceanics » waters** | | | | | | | | |
| January | 0,213 ± 0,042 | 2,8 ± 0,5 | 0,9 ± 0,5 | 2,7 ± 1,0 | 14 ± 3 | 19 ± 3 | 1,2 ± 0,3 | 27 - 46 |
| Start of March | 0,034 ± 0,027 | 0,9 ± 1,2 | 0,1 ± 0,1 | 1,0 ± 0,7 | > 100 | > 100 | 0,8 ± 0,8 | 40 |
| June | < 0,020 | 0,3 ± 0,3 | 0,2 ± 0,2 | 1,6 ± 0,5 | - | - | 0,3 ± 0,3 | 18 |
| **« girondines » waters** | | | | | | | | |
| End February | 0,289 ± 0,236 | 9,8 ± 4,1 | 0,5 ± 0,3 | 6,2 ± 2,2 | 51 ± 33 | 55 ± 36 | 1,6 ± 0,4 | 19 - 21 |
| End April | < 0,020 | 4,3 ± 2,1 | 0,7 ± 0,3 | 1,9 ± 1,2 | > 100 | > 100 | 3,5 ± 3,1 | 45 |
| End May | < 0,020 | 6,6 ± 5,9 | 0,9 ± 0,7 | 3,2 ± 4,6 | > 100 | > 100 | 16 ± 30 | 18 - 22 |
| End June | < 0,020 | 1,3 ± 1,6 | 0,2 ± 0,1 | 3,3 ± 1,7 | - | - | 0,3 ± 0,3 | 10 - 26 |
| Mid July | < 0,020 | 0,5 ± 0,9 | 0,5 ± 0,3 | 3,6 ± 2,1 | - | - | 0,1 ± 0,2 | 32 |
| Start October | 0,284 ± 0,062 | 4,9 ± 2,5 | 1,1 ± 0,2 | 6,6 ± 1,7 | 17 ± 7 | 23 ± 7 | 0,7 ± 0,2 | 12 |
| **« oceanics » waters** | | | | | | | | |
| End February | 0,127 ± 0,041 | 3,4 ± 1,0 | 0,4 ± 0,2 | 2,6 ± 0,9 | 30 ± 16 | 35 ± 18 | 1,5 ± 0,9 | 25 - 27 |
| End April | < 0,020 | 1,8 ± 0,6 | 0,7 ± 0,1 | 0,8 ± 0,3 | > 100 | > 100 | 2,7 ± 1,4 | 4 |
| End May | < 0,020 | 1,0 ± 0,3 | 1,0 ± 0,3 | 2,0 ± 1,0 | > 100 | > 100 | 0,6 ± 0,4 | 2 |
| Start October | 0,022 ± 0,031 | 0,2 ± 0,2 | 0,9 ± 0,3 | 1,6 ± 0,7 | - | - | 0,1 ± 0,1 | 43 |

*(Years: 1998 for the first group of sections, 1999 for the second group.)*

Table 1: Nutrients concentrations within the Gironde plume in late winter (from Labry et al. 2001)

[Figure]

Figure 2: Box-plot representation of nutrient concentrations (DIN, PO43- and Si(OH)4) between January and March (day 0 to day 69) measured (a) at the REPHY West Loscolo station (Bay of Vilaine) during the period 2011 -2019, (b) at the SOMLIT Ste Anne station (Bay of Brest) during the period 2001-2019

*Del Amo Y., Brzezinski M.A. (1999) The chemical form of dissolved Si taken up by marine diatoms, Journal of Phycology, 35, 1162-1170.*

*Gohin Francis, Lampert Luis, Guillaud Jean-Francois, Herbland Alain, Nezan Elisabeth (2003). Satellite and in situ observations of a late winter phytoplankton bloom, in the northern Bay of Biscay. Continental Shelf Research, 23(11-13), 1117-1141. Publisher's official version : https://doi.org/10.1016/S0278-4343(03)00088-8*

*Labry Claire, Herbland Alain, Delmas Daniel, Laborde P, Lazure Pascal, Froidefond J, Jegou Anne-Marie, Sautour B (2001). Initiation of winter phytoplankton blooms within the Gironde plume waters in the Bay of Biscay. Marine Ecology Progress Series, 212, 117-130.*

Here, the studied areas (Bay of Brest and Bay of Vilaine) are eutrophied bays with high nutrient concentrations. In our coastal waters, silicates are consumed first, then phosphates, then nitrates, but this occurs at the end of the first spring bloom, not at the onset. This is why our study focuses only on the physical conditions responsible for the IPGP. We clarify the information of a non-limitation of nutrients more clearly in our study by adding them for example in the Table 4 of the manuscript with the other environmental parameters and also in some parts in the text:

Abstract - "coastal temperate ecosystems under the influence of rivers highly rich in nutrients"

Introduction - "The river influence induces waters highly rich in nutrients."

Results part 4.1 - "However, at the beginning of the phytoplankton growing period (IPGP), the system is not nutrient limited in terms of nitrate, phosphorus and silicates."

There is in the understanding of these late winter blooms (between day 50 and 90) a critical issue for identifying the "major cause" of the "major disturbance" of the biological environment over the continental shelf of the Bay of Biscay. For the initiation of early offshore blooms, the light is the prevailing factor, not the SST; hence a verification of the critical depth hypothesis as formalized by Sverdrup.

Indeed, we agree with the referee that the light is one of the prevailing factors in the initiation of phytoplankton growth. However, in our system, the sea temperature is also a prerequisite of the phytoplankton growth and cold water temperature avoids earlier blooms in the season.

The critical depth hypothesis formalized by Sverdrup (1953) is based on the fact that phytoplankton blooms occur when surface mixing shoals to a depth shallower than a critical depth. In our studied region, the ecosystem does not evolve with mixed layer dynamics as observed in deeper environments (i.e. deep mixed layer depth in winter mainly due to wind forcings and shallower mixed layer depth in spring linked with the onset of stratification and weakening of wind induced mixing). Indeed, shallow waters(< 30m depth) in both bays are permanently vertically mixed mainly by the tides and the intensity of the mixing mainly fluctuates with tidal amplitude and wind intensity. The vertical stratification only occurs on a thin surface layer due to river runoffs in those bays for short time scales (few hours to few days during a flood event for example).

To avoid reader confusion, we rephrased the introduction of the manuscript to describe the local factors driving the phytoplankton growth dynamics.

Although criticized with real arguments, this simple theory may be locally verified in the bay of Biscay. The blooms occur in the clear, relatively cold, and stratified waters of the outer river plumes (Loire, Vilaine, Gironde). These blooms, sometimes very strong, have a high impact in biology as they provide foods for benthos at the end of winter and they consume a large part of the phosphorus stock in the surface layer with consequences in the phytoplankton size. As the concentration of Phosphorus in the rivers has been declining at high pace for these last twenty years, these blooms could have a stronger impact in the future.

Despite the decrease in winter phosphate concentrations in rivers (Ratmaya 2019), phosphate concentrations measured in the Bay of Vilaine and in the Bay of Brest before the IPGP date (day 68 = median day of the IPGP) are still high (Figure 2) and are not limiting phytoplankton growth (Ks = 0.09 µmol/L, Labry et al. 2001). We add the concentrations of nutrients in Table 2 of the manuscript.

My feeling is therefore that this study presents some interesting results but they have to be considered as a very local representation of much larger dynamics.

One of the aims of the project was to connect local phytoplankton dynamics with larger scale dynamics. However, in those shallow environments (<30m), the ecosystem dynamics is driven by a combination of local factors under influence of continental (rivers) and atmospheric forcings. The phytoplankton growth dynamics is then independent of large scale blooms observed in the bay of Biscay or the Iroise sea.

Considering these time series at the stations together with satellite data of chlorophyll-a and a 2 or 3-D model appear to be the next steps to propose for future investigations.

Thank you for this comment, we totally agree. We detailed the perspectives of this work in the new manuscript in the second paragraph of part 5 conclusions. Our first approach was based on *in situ* observations and a simplified 1DV modeling but we are indeed planning to extend our analyses to a 3D modeling approach for future investigations as well as satellite data.

A better consideration of the atmospheric environment would also benefit the understanding of these late winter blooms as anticyclonic conditions associated to high solar irradiance and low wind (hence lower turbidity) generally prevail at the onset of the late winter blooms in the Bay of Biscay.

We thank you for this comment. In our study, we concentrate on constraining components from the atmosphere for the ecosystem (solar irradiance, wind intensity). However, we explored the atmospheric conditions by looking at the atmospheric pressure time series (Figure 3). However, in our case, for both bays, the atmospheric conditions are not the same at each IPGP. The IPGP can occur during low pressure conditions (e.g. 2019) or during high pressure conditions (e.g. 2012) as for example in the Bay of Brest (Figure 3). We then consider the temperature and wind separately rather than considering the atmospheric pressure.

[Figure]

[Figure]

Figure 3: Two examples of atmospheric pressure time series (from AROME-Meteo-France model simulations) in the Bay of Brest with the date of IPGP for each year (black vertical line) and the median date of IPGP (day 69, black dashed vertical line) and the threshold of cyclonic/anticyclonic conditions at 1013 hPa (red horizontal line).

Specific comments: My general comments have implications in the abstract, the introduction, the discussion and the conclusion.

Abstract: "The use of a one-dimensional vertical model coupling hydrodynamics, biogeochemistry and sediment dynamics shows that the IPGP is generally dependent on the interaction between several drivers. Interannual changes are therefore not associated with a unique driver (such as increasing sea surface temperature)." Nobody would dare say that temperature is the unique driver of the IGPB. "Extreme event also impact the IGPB". Obvious but how is it quantified in the text? Not useful mentioning it.

We agree and the sentences have been deleted and reworded as follows in the abstract:

"*In situ* observations and a one-dimensional vertical model coupling hydrodynamics, biogeochemistry, and sediment dynamics show that the IPGP generally depends on the interaction between several environmental factors. IPGP is mainly conditioned, at the local scale, by sea surface temperature and available light conditions, controlled by the turbidity of the system before first blooms. "

"In both bays, IPGP can be delayed by cold spells and flood events at the end of winter if these extreme events last several days."

Introduction: "Moreover, theories proposed for the open oceans are not relevant in coastal zones." Really?

Following your comment this sentence has been deleted from the introduction and the following sentence reworded as follows:

"Coastal waters remain highly dynamic and productive ecosystems at the interface between land and sea and are distinguished from the waters of the open sea (*e.g.* Gohin *et al.*, 2019; Liu *et al.*, 2019)."

Discussion Extreme events: "In coastal stratified regions (e.g. under the influence of river plumes), strong wind and tidal mixing can enhance the mixing and break down stratification. Such conditions can also enhance phytoplankton production (Joordens et al., 2001). During the IPGP, except during floods, both regions are weakly stratified and are then less sensitive to combined wind/tidal short events." Not useful. In fact the stratification acts positively for initiating blooms in coastal and open waters at the end of winter.

Thanks for the comment, this part has been changed to:

"In coastal stratified regions (e.g. under the influence of river plumes), strong wind and tidal mixing can enhance the mixing and break down stratification thus distributing phytoplankton (Joordens et al., 2021). During the IPGP, except during floods, both regions are weakly stratified and are then less sensitive to combined wind/tidal short events."

Conclusion: You could imagine something of higher ambition than adding horizontal advection in the model ...!

Indeed, we thought to change the second paragraph of the part 5 conclusions in the manuscript and develop our ideas in terms of perspectives of this study.

Figure 7: From the legend we are looking for the Chl curve. It would be better to change the legend for something similar to "Day at the IGPB and environmental drivers: . Illustrations in 2011, 2013 and 2014. ...."

Thank you for your remark, the updated legend has been modified in the manuscirpt and in the supplementary part like this:

"Figure 7: IPGP dates and environmental drivers: flow of the Aulne, Vilaine and Loire rivers, Sea Surface Temperature (SST), wind intensity, PAR, turbidity and sea level. Illustrations in 2011 for a mean IPGP date in (a) the Bay of Brest and (b) the Bay of Vilaine; in 2013 for an early IPGP date in (c) the Bay of Brest; in 2014 for a late IPGP date in (d) the Bay of Vilaine. The mean IPGP date of each bay is represented by a dotted black line and the IPGP date of the year is represented by a straight black line. Thresholds of each environmental driver are represented by grey vertical lines corresponding to the mean conditions calculated 30 days around the IPGP date. Grey areas are time periods favorable to IPGP."

---

## Author Comment (AC2)

Biogeosciences Discuss., referee comment RC2 https://doi.org/10.5194/bg-2022-86-RC2, 2022 © Author(s) 2022. This work is distributed under the Creative Commons Attribution 4.0 License.

**Comment on bg-2022-86**

Anonymous Referee #2

Referee comment on "Interannual variability of the initiation of the phytoplankton growing period in two French coastal ecosystems" by Coline Poppeschi et al., Biogeosciences Discuss., https://doi.org/10.5194/bg-2022-86-RC2, 2022

*Main manuscript modifications are highlighted in red.*

**Main findings of the study**

In this manuscript, the authors describe the interannual variability of the phytoplankton blooms contrasting two coastal eutrophic French bays. By using a combination of high-frequency in situ information (buoys) and simulation model (IDV) they attempted to identify main environmental drivers (climatic – hydrological) that modulate variations in observed and estimated parameters of the phytoplankton growth given some explanations of the role of water temperature and turbidity variables. Here a main time delay in triggering phytoplankton blooms was detected during the 2010-2020 period. The authors also pointed out the strong influence of "extreme events" such as cold spells and floods over phytoplankton blooms during winter.

Thank you for this comprehensive summary of our study. We appreciate your constructive comments taken into account below in our answers.

Although the authors do some interesting observational/modeling approaches with the data available, the manuscript is mostly presented as a description of the data, thus, their interpretations remain mostly speculative. I strongly suggest using additional information such as inorganic nutrients since both bays are defined as eutrophic systems.

We know that some studies have shown the importance of nutrients in the development of phytoplankton blooms (Labry et al., 2001; Del Amo and Brzezinski, 1999). However, our two temperate coastal bays differ in that they are not nutrient limited and this is the case for inorganic nutrients. We have represented the nutrient concentrations in Figure 1. In the Bay of Vilaine and in the Bay of Brest the nutrient concentrations remain high until late March. When Labry et al. (2001) observed phosphate concentrations close to the limit of detection of the analytical method (< 0.05 µmol/l) in February, the median phosphate concentration is equal to 0.83 µmol/L in the bay of Vilaine and to 0.43 µmol/L in the Bay of Brest (Figure 1). The median silicate concentration (Figure 1, respectively 38.1 and 8.1 µmol/L in the Bay of Vilaine and the Bay of Brest) is highly above the half-saturation constant required for their assimilation by diatoms (Ks = 2 µmol/L, Del Amo and Brzezinski 1999).

[Figure]

Figure 1: Box-plot representation of nutrient concentrations (DIN, PO43- and Si(OH)4) between January and March (day 0 to day 69) measured (a) at the REPHY West Loscolo station (bay of Vilaine) during the period 2011 -2019, (b) at the SOMLIT Ste Anne station (bay of Brest) during the period 2001-2019

*Del Amo Y., Brzezinski M.A. (1999) The chemical form of dissolved Si taken up by marine diatoms, Journal of Phycology, 35, 1162-1170.*

*Labry Claire, Herbland Alain, Delmas Daniel, Laborde P, Lazure Pascal, Froidefond J, Jegou Anne-Marie, Sautour B (2001). Initiation of winter phytoplankton blooms within the Gironde plume waters in the Bay of Biscay. Marine Ecology Progress Series, 212, 117-130.*

We agree that this information of non-limitation in nutrients was not clearly enough presented in the manuscript. We therefore decided to present it more clearly by adding:

Table 4: addition of nutrient concentrations

Abstract - "coastal temperate ecosystems under the influence of rivers highly rich in nutrients"

Introduction - "The river influence induces waters highly rich in nutrients."

Results part 4.1 - "However, at the beginning of the phytoplankton growing period (IPGP), the system is not nutrient limited in terms of nitrate, phosphorus and silicates."

The main objective is not clearly defined, it should be written as a major one;

We have therefore restated the objectives more clearly in the manuscript as follows:

In this study, we aim to better understand interannual local changes in the IPGP in coastal temperate ecosystems in the current context of global climate change over the last 20 years. We first detect and analyze the temporal variability of the IPGP and we then quantify how environmental forcings influence its dynamics. To detect and analyze IPGP in coastal

environments, we develop a method, combining high-frequency decadal in situ observations and modeling, based on a 1DV hydro-sedimentary and biogeochemical coupled numerical model. The potential impact of hydro-meteorological extreme events, such as cold waves, flood events and wind bursts, on the IPGP is then investigated.

there are statements very descriptive at the Results section with too many figures, hard to understand showing different years, etc.

We agree and we have lightened the result part by rephrasing sentences for a more fluent reading. We also rewrite some paragraphs like in the 3.3.1, 3.3.2 and 4.3 sections of the manuscript in order to make it more fluid. To illustrate our results, we decided to keep existing figures.

In addition, I strongly recommend that the authors should make a major effort to write a general hypothesis or conceptual model for a future version.

In the introduction and in the results, our general view of the studied system and controlling factors has been rephrased and more explicitly stated.

**Especific comments:**

1.- The first limitation that comes to mind is the low sampling effort carried out in Bay of Vilaine for the chlorophyll-a (as fluorescence) variable, with only the second period survey completed. Would be possible to fill the gap of the first period with satellite images? In Bay of Brest appears to be an increase trend during the second period and probably both bays may be affected by similar drivers.

We agree that this would be interesting but the temporal sampling of the satellite is not high and values are not necessarily comparable (i.e. larger errors in satellite observations than in in situ observations) or available (Figure below). Here, we use *in situ* data from buoy measurements, so we cannot extend our series using satellite observations. Indeed, we also thought of investigating the effect of larger scale forcings (as climate indices - as the North Atlantic Oscillation - NAO) related to the similarities between the two bays but it appeared that local forcings are controlling the dynamics of the IPGP.

Example of available satellite observations at IPGP dates identified in HF *in situ* observations
(warning: illustrations are L4 sea surface temperature build from different satellites - Saulquin et Gohin, 2010)
( not shown but satellite observations are not available for every IPGP )

VILAINE                                                    BREST

[Figure]

Gohin Francis (2021). Long-Term Surveillance and Monitoring of Natural Events in Coastal Waters. In Remote Detection and Maritime Pollution: Chemical Spill Studies. 2021. Stéphane Le Floch, Frédéric Muttin (Eds). Print ISBN:9781786306395 |Online ISBN:9781119801849

Saulquin, B., & Gohin, F. (2010). Mean seasonal cycle and evolution of the sea surface temperature from satellite and in situ data in the English Channel for the period 1986–2006. International Journal of Remote Sensing, 31(15), 4069-4093.

2.- Inorganic nutrients: Although both bay are classified as eutrophic areas, the manuscript does not present data on N and Si; Si:N ands Si:P are interesting ratios to explore in the near surface layer, especially for diatoms, a groups that needs silicic acid for the frustule

and for dinoflagellates which are associated to nitrogen sources. Species/functional groups could respond more to ratios than concentrations; for example, cryptophytes and dinoflagellates may respond better to N sources (i.e. nitrate, ammonia), whereas diatoms could respond better to silicic acid concentrations.

We agree and we insert nutrient concentrations in Table 4 of the article (measured before the median IGPG date,day 68, by SOMLIT and REPHY).

We did not include data concerning the N/P and Si/N ratios because these parameters do not show interesting variations before the IPGP date: their median values are respectively 90.0 and 0.6 in the Bay of Vilaine, and 44.6 and 0.4 in the Bay of Brest (measured at REPHY and SOMLIT stations). All the ratio values are superior to 35 and inferior to 0.9 in the Bay of Vilaine, and superior to 20 and inferior to 0.65 in the Bay of Brest. These high N/P ratios and low Si/N ratios, compared to Redfield ratios (16 and 1), are characteristic of ecosystems subject to high winter nitrogen fluxes. The phytoplankton growth is not limited by nutrients before the IGPG date in both ecosystems. The phytoplankton population is then dominated by diatoms such as Skeletonema spp. and Chaetoceros spp.

3.- Model: There are several not clear issues in the parametrization of the model (Table 1), mainly in the methodology section: some restrictions on some parameters could be explained in more detail; for example, the initial value of O for dinoflagellates at the Bay of Vilaine; the starting values for N and Si nutrients, are they coming from a observed data base ?

We thank you for this comment. Indeed it was a typing error. The initial concentration of dinoflagellates in the Bay of Vilaine is equal to 0.1 micromolN $L^{-1}$ and not to zero so we corrected it in Table 1 of the article.

| Parameters | Bay of Brest | Bay of Vilaine |
|---|---|---|
| Dissolved $O_2$ *(mg $L^{-1}$)* | 9 | 10 |
| Mesozooplankton *($\mu molN\ L^{-1}$)* | 0.05 | 0.1 |
| Microzooplankton *($\mu molN\ L^{-1}$)* | 0.05 | 0.05 |
| Dinoflagellates *($\mu molN\ L^{-1}$)* | 0.05 | 0.1 |
| Diatoms *($\mu molN\ L^{-1}$)* | 0.5 | 0.5 |
| Soluble reactive phosphorus *($\mu mol\ L^{-1}$)* | 0.5 | 0.8 |
| Silicic acid *($\mu mol\ L^{-1}$)* | 10 | 30 |
| Nitrate *($\mu mol\ L^{-1}$)* | 16 | 30 |
| Ammonium *($\mu mol\ L^{-1}$)* | 0.5 | 0.25 |
| Coarse sand *(g $L^{-1}$)* | 0 | 0 |
| Fine sand *(g $L^{-1}$)* | 0 | 0 |
| Mud *(g $L^{-1}$)* | 0.03 | 0.05 |

**Table 1 Initial conditions in the water column for the MARS-1DV model for the beginning of the simulation the 15th February**

Nutrient data like all the initial values come from the MARS-3D model which was validated and then brought to equilibrium (Plus *et al.*, 2021). We clarified this in section 2.3.2 by adding the citation.

*Plus Martin, Thouvenin Benedicte, Andrieux Francoise, Dufois Francois, Ratmaya Widya, Souchu Philippe (2021). Diagnostic étendu de l'eutrophisation (DIETE). Modélisation biogéochimique de la zone Vilaine-Loire avec prise en compte des processus sédimentaires.*

*Description du modèle Bloom (BiogeochemicaL cOastal Ocean Model). RST/LER/MPL/21.15. https://archimer.ifremer.fr/doc/00754/86567/*

4.- Functional groups: Authors stated that *Skeletonema* spp. and *Chaetoceros* spp. as dominant species during both periods; would be possible that the increase in fluorescence is due to changes in taxonomical groups? Since both bays are under the influence of nutrient loading (mainly nitrogen), would be possible that the toxic dinoflagellates dominance would be part of the seasonal succession or interannual variability? Any evidence of more frequent and intense Harmful Algal Blooms (HABs) during the annual cycle or during climatological-hydrological extreme events ?

      *Skeletonema* spp. and *Chaetoceros* spp. are the dominant species at the date of each annual IPGP in both ecosystems. It is generally at the end of the first bloom that a shift is observed in the phytoplankton population dominance (Si or Si+P limitation). This study is only based on the IPGP because the fluorescence parameter is not robust enough to be able to study the whole growing period and to be linked with the seasonal phytoplankton succession. Quantifying *in vivo* phytoplankton communities using spectral fluorescence (Escoffier *et al.*, 2015) is still complex because it depends on factors such as the total levels of Chl-*a* and certain physiological states, such as those associated with light acclimation and/or nutrient stress and the species compositions of mixed assemblages. Fluorescence is used here only as a proxy for chlorophyll-a. Some toxic phytoplankton species (*Dinophysis sp., Alexandrium sp., Pseudo-nitzchia sp*.) may appears before the IPGP date but these species are not dominant (thresholds for HAB alerts are quite low; 100 cell/L for Dinophysis, 5000 cell/L for Alexandrium, 100 000 cell/L for Pseudo-nitzchia).

*Nicolas Escoffier, Cecile Bernard, Sahima Hamlaoui, Alexis Groleau, Arnaud Catherine, Quantifying phytoplankton communities using spectral fluorescence: the effects of species composition and physiological state, Journal of Plankton Research, Volume 37, Issue 1, January/February 2015, Pages 233–247, https://doi.org/10.1093/plankt/fbu085*

5.- According to authors, Temperature and Turbidity were the main drivers (I should prefer to say factors) of the variability of phytoplankton growth, however there is poor mechanistic explanation for their effects: to the mixing/stratification process such as Sverdrup hypothesis? Turbidity could affect in both ways to phytoplankton growth: large concentrations of terrigenous particles could decrease light penetration or increase inorganic nutrients (N, P) flux adjacent coastal land.

      We agree with the referee and we developed in the manuscript the processes we consider behind "Temperature and Turbidity". The manuscript has been modified in the introduction, results and discussion to clearly state observed processes.

      Concerning the mixing/stratification processes, classical theories can not be directly applied in our case. Indeed, the critical depth hypothesis formalized by Sverdrup (1953) is based on the fact that phytoplankton blooms occur when surface mixing shoals to a depth shallower than a critical depth. In our studied region, the ecosystem does not evolve with mixed layer dynamics as observed in deeper environments (i.e. deep mixed layer depth in winter mainly due to wind forcings and shallower mixed layer depth in spring linked with the onset of stratification and weakening of wind induced mixing). Shallow waters(< 30m depth) in both bays are permanently vertically mixed mainly by the tides and the intensity of the mixing mainly fluctuates with tidal amplitude and wind intensity. The vertical stratification only occurs on a thin surface layer due to river runoffs in those bays for short time scales (few hours to few days during a flood event for example).

Concerning the turbidity, before the beginning of the growing period, the system is not nutrient limited (as explained above) then the sensitivity if the system is only related to the effect of turbidity on light penetration in the water column. This point has also been clarified in the manuscript (introduction, results and discussion).

Following the referee's suggestion, we also changed the word "drivers" to "factors" in the entire manuscript as we agree with this terminology.

---

## Author Response (AR3)

Biogeosciences Discuss., referee comment RC1 https://doi.org/10.5194/bg-2022-86-RC1, 2022

**Comment on bg-2022-86**

Anonymous Referee #1

Referee comment on "Interannual variability of the initiation of the phytoplankton growing period in two French coastal ecosystems" by Coline Poppeschi et al., Biogeosciences Discuss., https://doi.org/10.5194/bg-2022-86-RC1, 2022

*Main manuscript modifications are highlighted in red.*

In this manuscript the decadal years variability of the IPGP (Initiation of the Phytoplankton Growing Period) is analyzed at two coastal stations located in the northern (Iroise Sea) and eastern Bay of Biscay (Bay of Vilaine). The phytoplankton biomass is related to fluorescence measured by instrumented buoys. The sensitivity analysis for identifying the major causes of the variability is made through a 1-D biogeochemical model applied to the year 2015. The major result of this work is that, despite different environments, the IBGP days are very similar at the two stations. No significant trend in the IPGP is observed in the time series However the variability is high and seems to show an earlier IGBP in the middle of the period -around year 2010- (at about day 60 against day 90 for the beginning and the end of the period). The results are interesting but this manuscript presents some flaws.

We thank the reviewer for the constructive reviews. We considered each point below.

The data used are fluorescence-derived Chlorophyll-a, the identification of the factors influencing the IPGP is made through a model and the conclusions are evasive.

The conclusions have been rephrased to be more detailed and less evasive in the first paragraph of part 5 conclusions and in the abstract. We then describe main conclusions built on model results but also on *in situ* observations.

On the first point, auxiliary analysed Chl-a concentrations collected bimonthly (which is a large interval for this purpose) corroborate nevertheless the IBGP derived from fluorescence data. The main issue comes therefore from the discussion based on model results. The model is considered as perfect and the causes of the variability are discussed from its outputs. From the introduction to the conclusion and throughout the discussion the real issues of the IGBP have not been considered with sufficient care. Blooms at local stations in river plumes may or may not occur, the true question is what happens at large scale? What is the connection with the dynamics of phytoplankton in the whole area?

The aim of this study was to investigate the initiation of the IPGP in nearshore waters under the influence of rivers highly rich in nutrients. The bathymetry (< 30 m) and the hydrodynamics of the Bay of Vilaine (Mor Braz) and the Bay of Brest do not allow them to be compared at larger scale respectively to the Bay of Biscay and to the Iroise sea. The late winter phytoplankton blooms reported in the northern Bay of Biscay by Labry et al.

(2001) and Gohin et al. (2003) were observed respectively in isobaths 60-30 and 120-80 m. According to the satellite observations, the phytoplankton dynamic in nearshore of south Brittany (Gohin 2010; Gohin 2012; marc.ifremer.fr website) is clearly different from the rest of the Bay of Biscay while the difference between the Gironde plume and the Bay of Biscay is less true (Figure 1). The maps of the monthly mean chl-a concentrations, as well as the annual cycles of chl-a, also show that it is in the nearshore of the Bay of Biscay, and in particular in the plumes of the Loire and Vilaine rivers, that the concentrations are the highest. For all these reasons, it does not seem appropriate to extend spatially (at the scale of the continental shelf of the bay of Biscay) our results obtained at one nearshore HF instrumented station.

In the manuscript introduction with the objectives have been modified to explain the local scale addressed in the study. We also add a specific sentence in section 2.1 "coastal temperate ecosystems".

[Figure]

*Figure 1: Mean Chl-a concentrations in January/February during 2003-2010 (from Gohin, 2012)*

*Gohin Francis, Saulquin Bertrand, Bryere Philippe (2010). Atlas de la Température, de la concentration en Chlorophylle et de la Turbidité de surface du plateau continental français et de ses abords de l'Ouest européen.https://archimer.ifremer.fr/doc/00057/16840/*

*Gohin Francis (2012). Répartition spatio-temporelle de la chlorophylle a. Sous-région marine Golfe de Gascogne. Evaluation initiale DCSMM. MEDDE, AAMP, Ifremer , Ref. DCSMM/EI/EE/GDG/12/2012, 13p. https://archimer.ifremer.fr/doc/00329/44009/*

*Labry Claire, Herbland Alain, Delmas Daniel, Laborde P, Lazure Pascal, Froidefond J, Jegou Anne-Marie, Sautour B (2001). Initiation of winter phytoplankton blooms within the Gironde plume waters in the Bay of Biscay. Marine Ecology Progress Series, 212, 117-130.*

*Morin, P., Le Corre, P., Marty, Y., L'Helguen, S., 1991, Evolution printanière des éléments nutritifs et du phytoplancton sur le plateau continental armoricain (Europe du Nord-Ouest). Oceanologica Acta 14:263-279.*

We know since the end of the 90's that strong early blooms within the Gironde plume may consume a large part of the winter Phosphorus stock at the beginning of March or even earlier. This has been attested also by satellite observations at broader scale.

According to the results of Labry et al. (2001) - Table 1, we agree that the late winter bloom consumes a large part of the winter phosphorus stock at the beginning of March within the Gironde plume. The situation is completely different in the Bay of Vilaine and in the Bay of Brest where nutrient concentrations remain high until late March. When Labry et al. (2001) observed phosphate concentrations close to the limit of detection of the analytical method (< 0.05 µmol/l) in February, the median phosphate concentration is equal to 0.83 µmol/L in the Bay of Vilaine and to 0.43 µmol/L in the Bay of Brest (Figure 2). The median silicate concentration (Figure 2, respectively 38.1 and 8.1 µmol/L in the Bay of Vilaine and the Bay of Brest) is highly above the half-saturation constant required for their assimilation by diatoms (Ks = 2 µmol/L, Del Amo and Brzezinski 1999). The waters are also not limited by silicate before the IGPG date, which accredits the presence of large diatoms at the IPGP date and not a phytoplankton population dominated by small cells.

| | | $PO_4$ µM | $NO_3$ µM | $NH_4$ µM | $Si(OH)_4$ µM | $NO_3$:$PO_4$ at:at | $N_{min}$:$PO_4$ at:at | $NO_3$:Si at:at | *n* |
|---|---|---|---|---|---|---|---|---|---|
| **1998** | **Turbid plume** | | | | | | | | |
| | January | 0,586 ± 0,086 | 26,2 ± 3,7 | 1,7 ± 1,5 | 17,9 ± 3,3 | 47 ± 6 | 50 ± 8 | 1,4 ± 0,1 | *4 - 5* |
| | Start March | 0,215 ± 0,064 | 13,1 ± 1,5 | 0,2 ± 0,2 | 8,1 ± 2,2 | 67 ± 25 | 69 ± 26 | 1,7 ± 0,6 | *4* |
| | **« girondines » waters** | | | | | | | | |
| | January | 0,320 ± 0,087 | 9,9 ± 3,6 | 1,1 ± 1,0 | 7,5 ± 2,0 | 31 ± 9 | 36 ± 9 | 1,3 ± 0,2 | *17 - 19* |
| | Start March | < 0,020 | 4,9 ± 2,6 | 0,1 ± 0,2 | 1,9 ± 1,0 | > 100 | > 100 | 2,4 ± 1,1 | *20 - 22* |
| | June | < 0,020 | 0,8 ± 0,5 | 0,3 ± 0,2 | 2,3 ± 0,6 | - | - | 0,4 ± 0,4 | *21 - 23* |
| | **« oceanics » waters** | | | | | | | | |
| | January | 0,213 ± 0,042 | 2,8 ± 0,5 | 0,9 ± 0,5 | 2,7 ± 1,0 | 14 ± 3 | 19 ± 3 | 1,2 ± 0,3 | *27 - 46* |
| | Start of March | 0,034 ± 0,027 | 0,9 ± 1,2 | 0,1 ± 0,1 | 1,0 ± 0,7 | > 100 | > 100 | 0,8 ± 0,8 | *40* |
| | June | < 0,020 | 0,3 ± 0,3 | 0,2 ± 0,2 | 1,6 ± 0,5 | - | - | 0,3 ± 0,3 | *18* |
| **1999** | **« girondines » waters** | | | | | | | | |
| | End February | 0,289 ± 0,236 | 9,8 ± 4,1 | 0,5 ± 0,3 | 6,2 ± 2,2 | 51 ± 33 | 55 ± 36 | 1,6 ± 0,4 | *19 - 21* |
| | End April | < 0,020 | 4,3 ± 2,1 | 0,7 ± 0,3 | 1,9 ± 1,2 | > 100 | > 100 | 3,5 ± 3,1 | *45* |
| | End May | < 0,020 | 6,6 ± 5,9 | 0,9 ± 0,7 | 3,2 ± 4,6 | > 100 | > 100 | 16 ± 30 | *18 - 22* |
| | End June | < 0,020 | 1,3 ± 1,6 | 0,2 ± 0,1 | 3,3 ± 1,7 | - | - | 0,3 ± 0,3 | *10 - 26* |
| | Mid July | < 0,020 | 0,5 ± 0,9 | 0,5 ± 0,3 | 3,6 ± 2,1 | - | - | 0,1 ± 0,2 | *32* |
| | Start October | 0,284 ± 0,062 | 4,9 ± 2,5 | 1,1 ± 0,2 | 6,6 ± 1,7 | 17 ± 7 | 23 ± 7 | 0,7 ± 0,2 | *12* |
| | **« oceanics » waters** | | | | | | | | |
| | End February | 0,127 ± 0,041 | 3,4 ± 1,0 | 0,4 ± 0,2 | 2,6 ± 0,9 | 30 ± 16 | 35 ± 18 | 1,5 ± 0,9 | *25 - 27* |
| | End April | < 0,020 | 1,8 ± 0,6 | 0,7 ± 0,1 | 0,8 ± 0,3 | > 100 | > 100 | 2,7 ± 1,4 | *4* |
| | End May | < 0,020 | 1,0 ± 0,3 | 1,0 ± 0,3 | 2,0 ± 1,0 | > 100 | > 100 | 0,6 ± 0,4 | *2* |
| | Start October | 0,022 ± 0,031 | 0,2 ± 0,2 | 0,9 ± 0,3 | 1,6 ± 0,7 | - | - | 0,1 ± 0,1 | *43* |

Table 1: Nutrients concentrations within the Gironde plume in late winter (from Labry et al. 2001)

[Figure]

(a)

(b)

Figure 2: Box-plot representation of nutrient concentrations (DIN, PO43- and Si(OH)4) between January and March (day 0 to day 69) measured (a) at the REPHY West Loscolo station (Bay of Vilaine) during the period 2011 -2019, (b) at the SOMLIT Ste Anne station (Bay of Brest) during the period 2001-2019

*Del Amo Y., Brzezinski M.A. (1999) The chemical form of dissolved Si taken up by marine diatoms, Journal of Phycology, 35, 1162-1170.*

*Gohin Francis, Lampert Luis, Guillaud Jean-Francois, Herbland Alain, Nezan Elisabeth (2003). Satellite and in situ observations of a late winter phytoplankton bloom, in the northern Bay of Biscay. Continental Shelf Research, 23(11-13), 1117-1141. Publisher's official version : https://doi.org/10.1016/S0278-4343(03)00088-8*

*Labry Claire, Herbland Alain, Delmas Daniel, Laborde P, Lazure Pascal, Froidefond J, Jegou Anne-Marie, Sautour B (2001). Initiation of winter phytoplankton blooms within the Gironde plume waters in the Bay of Biscay. Marine Ecology Progress Series, 212, 117-130.*

Here, the studied areas (Bay of Brest and Bay of Vilaine) are eutrophied bays with high nutrient concentrations. In our coastal waters, silicates are consumed first, then phosphates, then nitrates, but this occurs at the end of the first spring bloom, not at the onset. This is why our study focuses only on the physical conditions responsible for the IPGP. We clarify the information of a non-limitation of nutrients more clearly in our study by adding them for example in the Table 4 of the manuscript with the other environmental parameters and also in some parts in the text:

Abstract - "coastal temperate ecosystems under the influence of rivers highly rich in nutrients"

Introduction - "The river influence induces waters highly rich in nutrients."

Results part 4.1 - “However, at the beginning of the phytoplankton growing period (IPGP), the system is not nutrient limited in terms of nitrate, phosphorus and silicates.”

There is in the understanding of these late winter blooms (between day 50 and 90) a critical issue for identifying the “major cause” of the “major disturbance” of the biological environment over the continental shelf of the Bay of Biscay. For the initiation of early offshore blooms, the light is the prevailing factor, not the SST; hence a verification of the critical depth hypothesis as formalized by Sverdrup.

Indeed, we agree with the referee that the light is one of the prevailing factors in the initiation of phytoplankton growth. However, in our system, the sea temperature is also a prerequisite of the phytoplankton growth and cold water temperature avoids earlier blooms in the season.

The critical depth hypothesis formalized by Sverdrup (1953) is based on the fact that phytoplankton blooms occur when surface mixing shoals to a depth shallower than a critical depth. In our studied region, the ecosystem does not evolve with mixed layer dynamics as observed in deeper environments (i.e. deep mixed layer depth in winter mainly due to wind forcings and shallower mixed layer depth in spring linked with the onset of stratification and weakening of wind induced mixing). Indeed, shallow waters(< 30m depth) in both bays are permanently vertically mixed mainly by the tides and the intensity of the mixing mainly fluctuates with tidal amplitude and wind intensity. The vertical stratification only occurs on a thin surface layer due to river runoffs in those bays for short time scales (few hours to few days during a flood event for example).

To avoid reader confusion, we rephrased the introduction of the manuscript to describe the local factors driving the phytoplankton growth dynamics.

Although criticized with real arguments, this simple theory may be locally verified in the bay of Biscay. The blooms occur in the clear, relatively cold, and stratified waters of the outer river plumes (Loire, Vilaine, Gironde). These blooms, sometimes very strong, have a high impact in biology as they provide foods for benthos at the end of winter and they consume a large part of the phosphorus stock in the surface layer with consequences in the phytoplankton size. As the concentration of Phosphorus in the rivers has been declining at high pace for these last twenty years, these blooms could have a stronger impact in the future.

Despite the decrease in winter phosphate concentrations in rivers (Ratmaya 2019), phosphate concentrations measured in the Bay of Vilaine and in the Bay of Brest before the IPGP date (day 68 = median day of the IPGP) are still high (Figure 2) and are not limiting phytoplankton growth (Ks = 0.09 µmol/L, Labry et al. 2001). We add the concentrations of nutrients in Table 2 of the manuscript.

My feeling is therefore that this study presents some interesting results but they have to be considered as a very local representation of much larger dynamics.

One of the aims of the project was to connect local phytoplankton dynamics with larger scale dynamics. However, in those shallow environments (<30m), the ecosystem dynamics is driven by a combination of local factors under influence of continental (rivers) and atmospheric forcings. The phytoplankton growth dynamics is then independent of large scale blooms observed in the bay of Biscay or the Iroise sea.

Considering these time series at the stations together with satellite data of chlorophyll-a and a 2 or 3-D model appear to be the next steps to propose for future investigations.

Thank you for this comment, we totally agree. We detailed the perspectives of this work in the new manuscript in the second paragraph of part 5 conclusions. Our first approach was based on *in situ* observations and a simplified 1DV modeling but we are indeed planning to extend our analyses to a 3D modeling approach for future investigations as well as satellite data.

A better consideration of the atmospheric environment would also benefit the understanding of these late winter blooms as anticyclonic conditions associated to high solar irradiance and low wind (hence lower turbidity) generally prevail at the onset of the late winter blooms in the Bay of Biscay.

We thank you for this comment. In our study, we concentrate on constraining components from the atmosphere for the ecosystem (solar irradiance, wind intensity). However, we explored the atmospheric conditions by looking at the atmospheric pressure time series (Figure 3). However, in our case, for both bays, the atmospheric conditions are not the same at each IPGP. The IPGP can occur during low pressure conditions (e.g. 2019) or during high pressure conditions (e.g. 2012) as for example in the Bay of Brest (Figure 3). We then consider the temperature and wind separately rather than considering the atmospheric pressure.

[Figure]

[Figure]

Figure 3: Two examples of atmospheric pressure time series (from AROME-Meteo-France model simulations) in the Bay of Brest with the date of IPGP for each year (black vertical line) and the median date of IPGP (day 69, black dashed vertical line) and the threshold of cyclonic/anticyclonic conditions at 1013 hPa (red horizontal line).

Specific comments: My general comments have implications in the abstract, the introduction, the discussion and the conclusion.

Abstract: "The use of a one-dimensional vertical model coupling hydrodynamics, biogeochemistry and sediment dynamics shows that the IPGP is generally dependent on the interaction between several drivers. Interannual changes are therefore not associated with a unique driver (such as increasing sea surface temperature)." Nobody would dare say that temperature is the unique driver of the IGPB. "Extreme event also impact the IGPB". Obvious but how is it quantified in the text? Not useful mentioning it.

We agree and the sentences have been deleted and reworded as follows in the abstract:

"*In situ* observations and a one-dimensional vertical model coupling hydrodynamics, biogeochemistry, and sediment dynamics show that the IPGP generally depends on the interaction between several environmental factors. IPGP is mainly conditioned, at the local scale, by sea surface temperature and available light conditions, controlled by the turbidity of the system before first blooms. "

"In both bays, IPGP can be delayed by cold spells and flood events at the end of winter if these extreme events last several days."

Introduction: "Moreover, theories proposed for the open oceans are not relevant in coastal zones." Really?

Following your comment this sentence has been deleted from the introduction and the following sentence reworded as follows:

"Coastal waters remain highly dynamic and productive ecosystems at the interface between land and sea and are distinguished from the waters of the open sea (*e.g.* Gohin *et al.*, 2019; Liu *et al.*, 2019)."

Discussion Extreme events: "In coastal stratified regions (e.g. under the influence of river plumes), strong wind and tidal mixing can enhance the mixing and break down stratification. Such conditions can also enhance phytoplankton production (Joordens et al., 2001). During the IPGP, except during floods, both regions are weakly stratified and are then less sensitive to combined wind/tidal short events." Not useful. In fact the stratification acts positively for initiating blooms in coastal and open waters at the end of winter.

Thanks for the comment, this part has been changed to:

"In coastal stratified regions (e.g. under the influence of river plumes), strong wind and tidal mixing can enhance the mixing and break down stratification thus distributing phytoplankton (Joordens et al., 2021). During the IPGP, except during floods, both regions are weakly stratified and are then less sensitive to combined wind/tidal short events."

Conclusion: You could imagine something of higher ambition than adding horizontal advection in the model ...!

Indeed, we thought to change the second paragraph of the part 5 conclusions in the manuscript and develop our ideas in terms of perspectives of this study.

Figure 7: From the legend we are looking for the Chl curve. It would be better to change the legend for something similar to "Day at the IGPB and environmental drivers: . Illustrations in 2011, 2013 and 2014. ...."

Thank you for your remark, the updated legend has been modified in the manuscirpt and in the supplementary part like this:

"Figure 7: IPGP dates and environmental drivers: flow of the Aulne, Vilaine and Loire rivers, Sea Surface Temperature (SST), wind intensity, PAR, turbidity and sea level. Illustrations in 2011 for a mean IPGP date in (a) the Bay of Brest and (b) the Bay of Vilaine; in 2013 for an early IPGP date in (c) the Bay of Brest; in 2014 for a late IPGP date in (d) the Bay of Vilaine. The mean IPGP date of each bay is represented by a dotted black line and the IPGP date of the year is represented by a straight black line. Thresholds of each environmental driver are represented by grey vertical lines corresponding to the mean conditions calculated 30 days around the IPGP date. Grey areas are time periods favorable to IPGP."

Biogeosciences Discuss., referee comment RC2 https://doi.org/10.5194/bg-2022-86-RC2, 2022

**Comment on bg-2022-86**

Anonymous Referee #2

Referee comment on "Interannual variability of the initiation of the phytoplankton growing period in two French coastal ecosystems" by Coline Poppeschi et al., Biogeosciences Discuss., https://doi.org/10.5194/bg-2022-86-RC2, 2022

*Main manuscript modifications are highlighted in red.*

**Main findings of the study**

In this manuscript, the authors describe the interannual variability of the phytoplankton blooms contrasting two coastal eutrophic French bays. By using a combination of high-frequency in situ information (buoys) and simulation model (IDV) they attempted to identify main environmental drivers (climatic – hydrological) that modulate variations in observed and estimated parameters of the phytoplankton growth given some explanations of the role of water temperature and turbidity variables. Here a main time delay in triggering phytoplankton blooms was detected during the 2010-2020 period. The authors also pointed out the strong influence of "extreme events" such as cold spells and floods over phytoplankton blooms during winter.

Thank you for this comprehensive summary of our study. We appreciate your constructive comments taken into account below in our answers.

Although the authors do some interesting observational/modeling approaches with the data available, the manuscript is mostly presented as a description of the data, thus, their interpretations remain mostly speculative. I strongly suggest using additional information such as inorganic nutrients since both bays are defined as eutrophic systems.

We know that some studies have shown the importance of nutrients in the development of phytoplankton blooms (Labry et al., 2001; Del Amo and Brzezinski, 1999). However, our two temperate coastal bays differ in that they are not nutrient limited and this is the case for inorganic nutrients. We have represented the nutrient concentrations in Figure 1. In the Bay of Vilaine and in the Bay of Brest the nutrient concentrations remain high until late March. When Labry et al. (2001) observed phosphate concentrations close to the limit of detection of the analytical method (< 0.05 µmol/l) in February, the median phosphate concentration is equal to 0.83 µmol/L in the bay of Vilaine and to 0.43 µmol/L in the Bay of Brest (Figure 1). The median silicate concentration (Figure 1, respectively 38.1 and 8.1 µmol/L in the Bay of Vilaine and the Bay of Brest) is highly above the half-saturation constant required for their assimilation by diatoms (Ks = 2 µmol/L, Del Amo and Brzezinski 1999).

[Figure]

Figure 1: Box-plot representation of nutrient concentrations (DIN, PO43- and Si(OH)4) between January and March (day 0 to day 69) measured (a) at the REPHY West Loscolo station (bay of Vilaine) during the period 2011 -2019, (b) at the SOMLIT Ste Anne station (bay of Brest) during the period 2001-2019

*Del Amo Y., Brzezinski M.A. (1999) The chemical form of dissolved Si taken up by marine diatoms, Journal of Phycology, 35, 1162-1170.*

*Labry Claire, Herbland Alain, Delmas Daniel, Laborde P, Lazure Pascal, Froidefond J, Jegou Anne-Marie, Sautour B (2001). Initiation of winter phytoplankton blooms within the Gironde plume waters in the Bay of Biscay. Marine Ecology Progress Series, 212, 117-130.*

We agree that this information of non-limitation in nutrients was not clearly enough presented in the manuscript. We therefore decided to present it more clearly by adding:

Table 4: addition of nutrient concentrations

Abstract - "coastal temperate ecosystems under the influence of rivers highly rich in nutrients"

Introduction - "The river influence induces waters highly rich in nutrients."

Results part 4.1 - "However, at the beginning of the phytoplankton growing period (IPGP), the system is not nutrient limited in terms of nitrate, phosphorus and silicates."

The main objective is not clearly defined, it should be written as a major one;

We have therefore restated the objectives more clearly in the manuscript as follows:

In this study, we aim to better understand interannual local changes in the IPGP in coastal temperate ecosystems in the current context of global climate change over the last 20 years. We first detect and analyze the temporal variability of the IPGP and we then quantify how environmental forcings influence its dynamics. To detect and analyze IPGP in coastal

environments, we develop a method, combining high-frequency decadal in situ observations and modeling, based on a 1DV hydro-sedimentary and biogeochemical coupled numerical model. The potential impact of hydro-meteorological extreme events, such as cold waves, flood events and wind bursts, on the IPGP is then investigated.

there are statements very descriptive at the Results section with too many figures, hard to understand showing different years, etc.

We agree and we have lightened the result part by rephrasing sentences for a more fluent reading. We also rewrite some paragraphs like in the 3.3.1, 3.3.2 and 4.3 sections of the manuscript in order to make it more fluid. To illustrate our results, we decided to keep existing figures.

In addition, I strongly recommend that the authors should make a major effort to write a general hypothesis or conceptual model for a future version.

In the introduction and in the results, our general view of the studied system and controlling factors has been rephrased and more explicitly stated.

**Especific comments:**

1.- The first limitation that comes to mind is the low sampling effort carried out in Bay of Vilaine for the chlorophyll-a (as fluorescence) variable, with only the second period survey completed. Would be possible to fill the gap of the first period with satellite images? In Bay of Brest appears to be an increase trend during the second period and probably both bays may be affected by similar drivers.

We agree that this would be interesting but the temporal sampling of the satellite is not high and values are not necessarily comparable (i.e. larger errors in satellite observations than in in situ observations) or available (Figure below). Here, we use *in situ* data from buoy measurements, so we cannot extend our series using satellite observations. Indeed, we also thought of investigating the effect of larger scale forcings (as climate indices - as the North Atlantic Oscillation - NAO) related to the similarities between the two bays but it appeared that local forcings are controlling the dynamics of the IPGP.

Example of available satellite observations at IPGP dates identified in HF *in situ* observations
(warning: illustrations are L4 sea surface temperature build from different satellites - Saulquin et Gohin, 2010)
*( not shown but satellite observations are not available for every IPGP )*

VILAINE BREST

[Figure]

*Gohin Francis (2021). Long-Term Surveillance and Monitoring of Natural Events in Coastal Waters. In Remote Detection and Maritime Pollution: Chemical Spill Studies. 2021. Stéphane Le Floch, Frédéric Muttin (Eds). Print ISBN:9781786306395 |Online ISBN:9781119801849*

*Saulquin, B., & Gohin, F. (2010). Mean seasonal cycle and evolution of the sea surface temperature from satellite and in situ data in the English Channel for the period 1986–2006. International Journal of Remote Sensing, 31(15), 4069-4093.*

2.- Inorganic nutrients: Although both bay are classified as eutrophic areas, the manuscript does not present data on N and Si; Si:N ands Si:P are interesting ratios to explore in the near surface layer, especially for diatoms, a groups that needs silicic acid for the frustule

and for dinoflagellates which are associated to nitrogen sources. Species/functional groups could respond more to ratios than concentrations; for example, cryptophythes and dinoflagellates may respond better to N sources (i.e. nitrate, ammonia), whereas diatoms could respond better to silicic acid concentrations.

We agree and we insert nutrient concentrations in Table 4 of the article (measured before the median IGPG date,day 68, by SOMLIT and REPHY).

We did not include data concerning the N/P and Si/N ratios because these parameters do not show interesting variations before the IPGP date: their median values are respectively 90.0 and 0.6 in the Bay of Vilaine, and 44.6 and 0.4 in the Bay of Brest (measured at REPHY and SOMLIT stations). All the ratio values are superior to 35 and inferior to 0.9 in the Bay of Vilaine, and superior to 20 and inferior to 0.65 in the Bay of Brest. These high N/P ratios and low Si/N ratios, compared to Redfield ratios (16 and 1), are characteristic of ecosystems subject to high winter nitrogen fluxes. The phytoplankton growth is not limited by nutrients before the IGPG date in both ecosystems. The phytoplankton population is then dominated by diatoms such as Skeletonema spp. and Chaetoceros spp.

3.- Model: There are several not clear issues in the parametrization of the model (Table 1), mainly in the methodology section: some restrictions on some parameters could be explained in more detail; for example, the initial value of O for dinoflagellates at the Bay of Vilaine; the starting values for N and Si nutrients, are they coming from a observed data base ?

We thank you for this comment. Indeed it was a typing error. The initial concentration of dinoflagellates in the Bay of Vilaine is equal to 0.1 micromolN $L^{-1}$ and not to zero so we corrected it in Table 1 of the article.

| Parameters | Bay of Brest | Bay of Vilaine |
|---|---|---|
| Dissolved $O_2$ *(mg $L^{-1}$)* | 9 | 10 |
| Mesozooplankton *(µmolN $L^{-1}$)* | 0.05 | 0.1 |
| Microzooplankton *(µmolN $L^{-1}$)* | 0.05 | 0.05 |
| Dinoflagellates *(µmolN $L^{-1}$)* | 0.05 | 0.1 |
| Diatoms *(µmolN $L^{-1}$)* | 0.5 | 0.5 |
| Soluble reactive phosphorus *(µmol $L^{-1}$)* | 0.5 | 0.8 |
| Silicic acid *(µmol $L^{-1}$)* | 10 | 30 |
| Nitrate *(µmol $L^{-1}$)* | 16 | 30 |
| Ammonium *(µmol $L^{-1}$)* | 0.5 | 0.25 |
| Coarse sand *(g $L^{-1}$)* | 0 | 0 |
| Fine sand *(g $L^{-1}$)* | 0 | 0 |
| Mud *(g $L^{-1}$)* | 0.03 | 0.05 |

**Table 1 Initial conditions in the water column for the MARS-1DV model for the beginning of the simulation the 15th February**

Nutrient data like all the initial values come from the MARS-3D model which was validated and then brought to equilibrium (Plus *et al.*, 2021). We clarified this in section 2.3.2 by adding the citation.

*Plus Martin, Thouvenin Benedicte, Andrieux Francoise, Dufois Francois, Ratmaya Widya, Souchu Philippe (2021). Diagnostic étendu de l'eutrophisation (DIETE). Modélisation biogéochimique de la zone Vilaine-Loire avec prise en compte des processus sédimentaires.*

*Description du modèle Bloom (BiogeochemicaL cOastal Ocean Model). RST/LER/MPL/21.15. https://archimer.ifremer.fr/doc/00754/86567/*

4.- Functional groups: Authors stated that *Skeletonema* spp. and *Chaetoceros* spp. as dominant species during both periods; would be possible that the increase in fluorescence is due to changes in taxonomical groups? Since both bays are under the influence of nutrient loading (mainly nitrogen), would be possible that the toxic dinoflagellates dominance would be part of the seasonal succession or interannual variability? Any evidence of more frequent and intense Harmful Algal Blooms (HABs) during the annual cycle or during climatological-hydrological extreme events ?

*Skeletonema* spp. and *Chaetoceros* spp. are the dominant species at the date of each annual IPGP in both ecosystems. It is generally at the end of the first bloom that a shift is observed in the phytoplankton population dominance (Si or Si+P limitation). This study is only based on the IPGP because the fluorescence parameter is not robust enough to be able to study the whole growing period and to be linked with the seasonal phytoplankton succession. Quantifying *in vivo* phytoplankton communities using spectral fluorescence (Escoffier *et al.*, 2015) is still complex because it depends on factors such as the total levels of Chl-*a* and certain physiological states, such as those associated with light acclimation and/or nutrient stress and the species compositions of mixed assemblages. Fluorescence is used here only as a proxy for chlorophyll-a. Some toxic phytoplankton species (*Dinophysis sp., Alexandrium sp., Pseudo-nitzchia sp.*) may appears before the IPGP date but these species are not dominant (thresholds for HAB alerts are quite low; 100 cell/L for Dinophysis, 5000 cell/L for Alexandrium, 100 000 cell/L for Pseudo-nitzchia).

*Nicolas Escoffier, Cecile Bernard, Sahima Hamlaoui, Alexis Groleau, Arnaud Catherine, Quantifying phytoplankton communities using spectral fluorescence: the effects of species composition and physiological state, Journal of Plankton Research, Volume 37, Issue 1, January/February 2015, Pages 233–247, https://doi.org/10.1093/plankt/fbu085*

5.- According to authors, Temperature and Turbidity were the main drivers (I should prefer to say factors) of the variability of phytoplankton growth, however there is poor mechanistic explanation for their effects: to the mixing/stratification process such as Sverdrup hypothesis? Turbidity could affect in both ways to phytoplankton growth: large concentrations of terrigenous particles could decrease light penetration or increase inorganic nutrients (N, P) flux adjacent coastal land.

We agree with the referee and we developed in the manuscript the processes we consider behind "Temperature and Turbidity". The manuscript has been modified in the introduction, results and discussion to clearly state observed processes.

Concerning the mixing/stratification processes, classical theories can not be directly applied in our case. Indeed, the critical depth hypothesis formalized by Sverdrup (1953) is based on the fact that phytoplankton blooms occur when surface mixing shoals to a depth shallower than a critical depth. In our studied region, the ecosystem does not evolve with mixed layer dynamics as observed in deeper environments (i.e. deep mixed layer depth in winter mainly due to wind forcings and shallower mixed layer depth in spring linked with the onset of stratification and weakening of wind induced mixing). Shallow waters(< 30m depth) in both bays are permanently vertically mixed mainly by the tides and the intensity of the mixing mainly fluctuates with tidal amplitude and wind intensity. The vertical stratification only occurs on a thin surface layer due to river runoffs in those bays for short time scales (few hours to few days during a flood event for example).

Concerning the turbidity, before the beginning of the growing period, the system is not nutrient limited (as explained above) then the sensitivity if the system is only related to the effect of turbidity on light penetration in the water column. This point has also been clarified in the manuscript (introduction, results and discussion).

Following the referee's suggestion, we also changed the word “drivers” to “factors” in the entire manuscript as we agree with this terminology.

**Report #1**

accepted but with better editing work

We thank the reviewer and we agreed that this article can be improved. All authors read it again with great attention to the English, and the writing and modifications have been taken into account.

*Main manuscript modifications are highlighted in red.*

Report #2

I have not seen the first version of this manuscript. The rebuttal letter and changes to the text show – in my opinion – that the remarks of the reviewers have been sufficiently addressed. Nevertheless, some responses can still be improved. I also have some additional remarks (see below), but I consider these to be minor remarks.

We thank the reviewer for his/her constructive review. We considered each point below.

*Main manuscript modifications are highlighted in red.*

Referee 1 asked how these local changes are connected to the dynamics in the whole area (Bay of Biscay). While it is now clearly stated in the objectives that the study concerns local changes in IPGP, it is still not clear how your findings relate to variability in the offshore (open ocean/shelf) environment. I believe that the dynamics in the Bay of Biscay are well-described, so maybe a short description of this can be given in the introduction, and it can be even more clearly stated in the discussion how the local dynamics relate (rather absence of relation) to the offshore dynamics.

To consider this comment, we better described the dynamics in the Bay of Biscay and how local processes are related to the offshore dynamics in the "Study area" section:

Our study focuses on two northwestern French coastal temperate ecosystems located in the Bay of Biscay, the Bay of Brest and the Bay of Vilaine, two ecosystems impacted by excessive nutrient inputs from watersheds, but exposed to different hydrodynamic conditions.

The Bay of Biscay is a region with a complex system of coastal currents influenced by the combined effects of seasonal wind regimes and important river discharges modulated by large-scale gyre circulation patterns (Ferrer *et al*., 2009; Lazure and Jégou, 1998; Lazure *et al*., 2006; Isemer and Hasse, 1985; Pingree and Le Cann, 1989, 1990; Le Boyer *et al*., 2013; Lazure *et al*., 2006; Charria *et al*., 2013). In the Iroise Sea, at spring tide close to the islands and capes, tidal currents can reach 4 m $s^{-1}$ (Muller *et al*., 2010). This tidal circulation combined with meteorological forcings and sharp thermal gradients generate a strongly variable local circulation. In the vicinity of the Loire estuary, the freshwater discharges in the surface layers induce important density gradients driving a poleward circulation (about 10 cm $s^{-1}$) modulated by wind forcings (Lazure and Jégou, 1998; Lazure *et al*., 2006). The river plumes can propagate under specific conditions towards the South-West.

In the discussion, we introduce those sentences to more clearly state how the local dynamics relates to the offshore dynamics:

The IPGP appears to be more controlled by local environmental drivers than by regional environmental drivers, the IPGP being earlier in one site than in the other during half of the studied years: for example, the 2012 IPGP is early in the Bay of Vilaine (day 53), but late in the Bay of Brest (day 80), related to strong wind activity and low PAR on the last bay. The offshore regional dynamics will induce limited impacts on local hydrodynamical features that will change IPGP.

In the rebuttal to Referee 2, it is argued that satellite images are not available (with reference to a figure below, on p. 12 of rebuttal). But on the basis of this figure, I would conclude that suitable images are indeed available, as the images show clear weather conditions and the two coastal bays seem to harbour enough high-resolution signal.

While we agree that satellite data are available for this region, we would also like to point out that the spatio-temporal resolution of the currently available products (even if we can catch interesting patterns, they remain truncated due to cloud coverage. Furthermore, the temporal resolution remains low) is not suitable for our study, and more specifically for the study of *in situ* data at high acquisition frequency. Therefore, we do not prefer to add satellite data that we do not find relevant to the issue addressed in our paper, which would make our analyses more complex or even biased.

On p 13 of the rebuttal: is it not strange, given the low Si:N ratios in both systems, that diatoms then dominate the phytoplankton? (cf the argument of the reviewer)

The Si:N ratios in both systems are indeed quite low before the IPGP. But, this ratio, combined with a high N:P ratio, is characteristic of ecosystems subject to high winter nitrogen fluxes. As explained p. 9 of the rebuttal, the median silicate concentration (Figure 1 of the rebuttal, respectively 38.1 and 8.1 µmol/L in the Bay of Vilaine and the Bay of Brest) is highly above the half-saturation constant required for their assimilation by diatoms (Ks = 2 µmol/L, Del Amo and Brzezinski 1999). The phytoplankton growth is neither limited by the orthophosphate concentration (median concentration respectively equal to 0,8 and 0,4 µmol/L in the Bay of Vilaine and the Bay of Brest). It is therefore not surprising to have a phytoplankton population dominated by diatoms before the IPGP. The nutrient concentrations have been added in Table 4.

In addition, I have also read through the revised manuscript and have some additional comments. I think that overall the language could be improved. In some instances, it is not clear to me what is meant, and these parts should be rephrased:

To consider this comment, the revised version of the paper has been carefully revised to improve the grammar and readability. As suggested by Reviewer 3, we rewrote the following sentences:

• Line 15 – '… the effect of climate-induced changes on…'

We modified the sentence, as follows: "... climate-induced impacts on …"

• Lines 19-21 – this part of the sentence is confusing: '…available light depending from solar radiation, chlorophyll concentration and turbidity - sea temperature - turbulence driven by currents, wind direction and intensity and tidal mixing - nutrients from river flow.' Why use the hyphens? It is also not clear to me why chlorophyll a is listed as an environmental factor potentially influencing variation in IPGP, as chl a is part of the IPGP.

The referee is right. In the revised version of the manuscript, we now say: "In both coastal ecosystems, we observed a large interannual variation in IPGP influenced by sea temperature,

river inputs, light availability (modulated by solar radiation and water turbidity), and turbulent mixing generated by tidal currents, wind stress and river runoff."

• Line 37 – what do you mean with 'determined with specific scales'?

To clarify this sentence, we modified: "No consensus emerges among these hypotheses - especially because most of these concepts have been defined at specific temporal and spatial scales (Caracciolo *et a*l., 2021; Chiswell *et al*., 2015) - and the debate is still open, in particular due to the use of more efficient models, the availability of new observations, and the ensuing collection of large *in situ* datasets (Boss and Behrenfeld, 2010; Rumyantseva *et al*., 2019)."

• Line 50: 'The variability of IPGP' ◊ 'Variation in IPGP …'

We agree with the reviewer. We directly replaced: "Temporal variation in IPGP [...]".

• Line 59 – what do you mean with 'land-based transfers'? Transfers of what?,

We deleted this sentence because it was repetitive with the sentence just before "continental erosion".

• Line 94-95 – how does the macrotidal regime prevent the formation of green tides or the shift from diatoms to non-siliceous plankton?

We thank the reviewer for this comment. We now better explain this part in the revised version of the manuscript. We now say: "Due to the macrotidal regime, associated with a strong vertical mixing, the high nitrate concentrations do not generate important green tides (Le Pape *et al*., 1997). Strong decreases in the Si:N and Si:P ratios did not exhibit dramatic phytoplankton community shifts from diatoms to non-siliceous species in spring (Del Amo *et al*., 1997) because of the high Si recycling (Ragueneau *et al*., 2002; Beucher *et al*, 2004)."

• Line 96 – 'according to' = because of ?

We changed, as follows: " [...] because of [...]".

• Line 120 – 'every 20 and 60 minutes' – do you mean that one buoy measures every 20 mins and the other one every hour

This is correct. We clarified the sentence in the revision: "Environmental parameters (SST, salinity, turbidity, dissolved oxygen and Chl-*a* fluorescence) are measured at 1 to 2 m below the surface every 20 minutes (COAST-HF-Iroise) or every hour (COAST-HF-Molit)."

• Lines 216-217 – I don't understand what you mean with 'too late IPGP', 'too early IPGP', …? Please rephrase.

We modify this sentence by: "[...] too late (method 1) or a too early (method 2) detection [...]".

• Lines 359-360 – 'In the Bay of Brest…' I don't understand this sentence nor how it is related to the previous sentence.

The sentence is now: "Due to the most unfavorable conditions, the IPGP occurs 9 days and 64 days later in the Bay of  Brest and the Bay of Vilaine, respectively."

• Line 549 – what do you mean with 'distributing phytoplankton'? Vertical mixing and disruption of chlorophyll maxima?

To be clearer, we modified: " [...] does not favor phytoplankton growth".

In the introduction you mention three main theories to explain how blooms initiate. Would it be possible to very briefly explain these three hypotheses and how they differ? Also, you don't come back to these in the discussion. Is this because these are not relevant in the context of shallow, well-mixed coastal systems under the influence of river plumes? It may be good to briefly address this issue.

We agree and we take into account the comment from the reviewer by explaining the three hypothesis in the introduction part and also by considering back those theories in the discussion part:

In the 'Introduction' section, we explained: "For Sverdrup (1953), phytoplankton blooms occur when surface mixed layer shoals to a depth shallower than the critical depth, according to light conditions. While Huisman *et al*. (1999) agreed with Sverdrup (1953), he proposed that relaxation of turbulent mixing allows bloom to develop if it occurs below a critical turbulence rate. Behrenfel (2010) observed blooms occurring in the absence of spring mixed layer shoaling, and declared that the initiation of bloom is controlled by a balance between phytoplankton growth and grazing rate, and suggested a seasonal control of this balance by physical processes."

In the 'Discussion' section, we addressed the issue as follows: "The main theories to explain the initiation of phytoplankton blooms (Sverdrup, 1953; Huisman *et al*., 1999; Banse, 1994) are not relevant in the context of shallow, well-mixed coastal waters under the influence of river plumes. In our studied region, the ecosystem does not evolve with mixed layer dynamics, as observed in deeper environments. Both bays are permanently vertically mixed mainly by tides, and vertical stratification only occurs on a thin surface layer due to river runoffs at short time scales."